# SENSE: SENsing Similarity SEeing Structure

## Abstract

Low-dimensional embeddings are central to analyzing and visualizing high-dimensional data. However, widely adopted NE methods assume centralized access to all data an unrealistic constraint in privacy-sensitive, decentralized environments. We propose **SENSE**, a geometry-aware, privacy-preserving framework for global neighbor embedding without raw data exchange. SENSE reconstructs global structure using local distance measurements and structured matrix completion, enabling embeddings that preserve both local and global geometry in Euclidean and hyperbolic spaces. It further integrates contrastive learning by deriving cross-client positive and negative pairs from estimated similarities, effectively generalizing negative sampling under structural constraints. Experiments across diverse real-world datasets show that SENSE achieves embedding quality on par with centralized baselines, while offering strong privacy guarantees. Theoretical analysis provides formal bounds on reconstruction fidelity and privacy, establishing conditions under which structure and confidentiality are jointly preserved. [1]

## 1 Introduction

Neighbor embedding (NE) methods are widely used for dimensionality reduction (DR), enabling interpretable low-dimensional visualizations of high-dimensional data [51]. Techniques like t-SNE [53], UMAP [37], MDS [15], and PHATE [38] are effective for visualization [9], anomaly detection [46], and exploratory analysis [16]. These methods, however, assume centralized access to complete pairwise similarity matrices an assumption often violated in real-world settings. In domains such as healthcare [45], finance [8], and mobile networks [34], data is distributed across clients and subject to strict privacy constraints. In such settings, standard NE methods fail due to the absence of global distance information especially problematic for attraction-repulsion frameworks like t-SNE and UMAP [6, 56] that depend on complete similarity graphs to balance local and global structure. Recent work links NE with contrastive learning [10, 11], further emphasizing the importance of accurate pairwise similarities. In privacy-constrained regimes, however, such structure is either missing or only partially available, making decentralized contrastive NE a challenging problem.

**Related Work.** Several approaches have been proposed to address this gap, but they fall short on scalability, privacy, or deployment realism. SMAP [57] offers strong privacy via encrypted multi-party computation, but its cryptographic overhead renders it impractical for large-scale use. FedNE [33] introduces a federated NE framework but lacks intrinsic privacy guarantees and incurs repeated server-client interactions, making it communication heavy. Methods like dSNE [48] and FdSNE [47] require full shared reference datasets for alignment, an unrealistic assumption in many settings, and diverge from standard FL protocols while also introducing high communication and privacy costs. More recently, MMD-based distribution alignment [43] has been used to generate synthetic shared data, but it assumes multi-sample clients and is fragile in single data sample per client scenarios common to IoT and mobile devices. Moreover, it risks adversarial corruption of synthesized distributions and introduces additional computational burden. To address these limitations, we propose **SENSE**,

---

[1]Code is available at SENSE

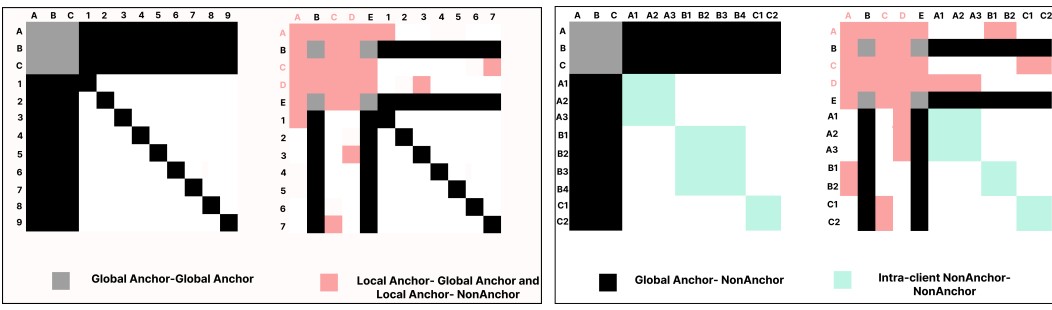

Figure 1: Observed entries in the global distance matrix $D$ under four SENSE configurations: (1) *Pointwise-Full*, (2) *Pointwise-Partial*, (3) *Multisite-Full*, and (4) *Multisite-Partial*. These differ in the visibility of Anchor–NonAnchor (A–NA) and NA–NA blocks, governed by client-level data locality and anchor access. *Multisite* settings permit intra-client NA–NA observations (e.g., A1, A2, ..., C2), while *Pointwise* settings restrict each client to a single NA (e.g., 1, 2, ..., 9). *Full* modes provide all NAs with access to the global anchor set (e.g., A–E), yielding complete A–NA blocks; *Partial* modes expose disjoint anchor subsets per client, resulting in sparse and structured observations.

a unified, geometry-aware framework for *privacy-preserving decentralized neighbor embedding*. SENSE supports both Euclidean and hyperbolic geometries the latter being critical for embedding hierarchical structures in social and biological data [30, 36]. Unlike prior work, SENSE reconstructs global structure from sparse local distance observations using anchor-based measurements, without requiring raw data sharing, iterative communication, or centralized storage. The completed distance matrix is then used with classical NE methods, contrastive NE, and hyperbolic CoSNE [22].

Although anchor sharing is sometimes perceived as a constraint in decentralized settings [43], it serves as a robust, principled, and privacy-preserving coordination mechanism increasingly adopted in practice. When curated by a trusted server, anchors can be synthetic, anonymized, or sourced from public data completely decoupled from private client records. This mitigates leakage risks inherent to client-generated anchors, which are vulnerable to reconstruction or membership inference, especially in small or skewed-client regimes [43]. Server-curated anchors offer stability, auditability, and adversarial robustness, enabling secure global coordination without compromising privacy. This paradigm is already in use across real-world systems in healthcare [7, 27], genomics [35, 44], finance [2], and mobile/NLP applications [23, 32], illustrating that carefully designed anchor-based schemes are both secure and essential for scalable decentralized learning. Motivated by this, we argue that anchors should be treated as core architectural components rather than ad hoc artifacts. SENSE leverages anchor-based coordination in conjunction with tools from distance matrix completion, network localization, and low-rank recovery, providing formal guarantees for reconstructing global geometry from partial observations. When combined with contrastive learning, it further enhances alignment and expressiveness, bridging classical and modern NE paradigms. SENSE introduces the following key innovations:

- *Privacy by design:* Estimates global structure using only local distance measurements, eliminating the need for encryption or differential privacy.
- *Communication-efficient and geometry-aware:* Requires a single server–client interaction, and supports both Euclidean and hyperbolic spaces for modeling flat and hierarchical data.
- *Deployment flexibility:* Operates under two regimes (Figure 1): *SENSE-Pointwise* for single-point clients (e.g., edge/mobile), and *SENSE-Multisite* for multi-sample clients (e.g., hospitals, banks).
- *Provable reliability:* Offers theoretical guarantees on both privacy preservation and embedding fidelity, validated across diverse modalities and geometries.

These properties make *SENSE* suitable for privacy-sensitive, structurally diverse domains. Hospitals can jointly visualize patient data without violating HIPAA/GDPR [50], banks can detect fraud patterns without sharing transactions [3], and mobile/IoT clients with a single sample can still contribute to global embeddings [4, 42]. Genomic labs can embed single-cell transcriptomes into a shared hyperbolic space that preserves cellular hierarchy and privacy [1, 52]. Crucially, *SENSE* also supports evolving data scenarios and dynamic client participation, new clients or data points can be integrated by estimating only their partial distances to a subset of existing entities, avoiding full re-computation and preserving global coherence with minimal overhead. This makes *SENSE* not only privacy-preserving and geometry-aware but also inherently scalable to dynamic and federated ecosystems.

## 2  Background and Problem Formulation.

**Neighbor Embedding (NE).**  Methods like t-SNE [53] and UMAP [10] embed high-dimensional data $\mathbf{X} = \{x_i\}_{i=1}^n \subset \mathbb{R}^{d_h}$ into a low-dimensional space $\mathbf{Y} = \{y_i\}_{i=1}^n \subset \mathbb{R}^{d_\ell}$ by preserving pairwise structure. These methods are distance-driven. They transform distances into similarities via kernels to preserve relational structure (see Appendix A.1, A.2). Let $D_{ij}^{d_h} = \|x_i - x_j\|$ and $D_{ij}^{d_\ell} = \|y_i - y_j\|$ denote distances in the high- and low-dimensional spaces. These are mapped to similarities via kernel functions: $S_{ij}^{d_h} = f(D_{ij}^{d_h}), \quad S_{ij}^{d_\ell} = g(D_{ij}^{d_\ell})$, where $f$ and $g$ are typically Gaussian, Laplacian, or Cauchy kernels. The general NE objective minimizes the divergence between the two similarity matrices:

$$\mathcal{L}(\mathbf{Y}) = \sum_{i,j} \mathcal{D}(S_{ij}^{d_h},\, S_{ij}^{d_\ell}), \tag{1}$$

where $\mathcal{D}$ is a divergence measure such as KL divergence or binary cross-entropy.

**Contrastive Neighbor Embedding.**  CNE [11] extends NE into the contrastive learning framework by training an encoder $f_\theta$ to map $\mathbf{x}_i$ to $\mathbf{y}_i = f_\theta(\mathbf{x}_i)$ such that the neighborhood structure from a $k$-NN graph is preserved. CNE uses a distance-aware contrastive loss (see Def A.3 in Appendix), framed as a binary similarity matching problem. Let $S^{d_h} \in \{0,1\}^{n \times n}$ denote ground-truth neighborhood indicators and $S^{d_l}$ denote kernel-based similarities in the embedding space. The loss is a weighted binary cross-entropy:

$$\mathcal{L}(\mathbf{Y}) = -\sum_{i,j} \left[ S_{ij}^{d_h} \log S_{ij}^{d_l} + b(1 - S_{ij}^{d_h}) \log(1 - S_{ij}^{d_l}) \right]. \tag{2}$$

*Key Challenges in Decentralized Settings.* **(C1)** CNE, like NE, relies on a full similarity matrix, which is unavailable in privacy-sensitive, decentralized settings. **(C2)** Conventional distributed learning captures only intra-client structure, omitting crucial inter-client neighbor information. **(C3)** Clients lack access to global data, leading to incorrect kNN graphs and biased negative sampling, as true neighbors may reside on other clients.

**CO-SNE (for Hyperbolic Data).**  Hierarchical structures in social, biological, and knowledge graphs grow exponentially, making Euclidean embeddings unsuitable due to distortion of tree-like geometry. Hyperbolic space, with constant negative curvature, naturally models such growth and supports hierarchy-aware learning [19, 36, 40] (see Appendix A.3.1). Standard methods like t-SNE assume Euclidean geometry and distort global structure when applied to hyperbolic data, collapsing depth and relative positioning. CO-SNE [22] extends t-SNE to hyperbolic space (see Def A.4). It preserves both local and global structure using distance-aware kernels in hyperbolic geometry: $S_{ij}^{d_h} = f(d_{\mathbb{B}^n}(x_i, x_j)), \quad S_{ij}^{d_l} = g(d_{\mathbb{B}^2}(y_i, y_j))$, where $f$ is a hyperbolic normal kernel and $g$ is a heavy-tailed hyperbolic Cauchy kernel. A regularization term also aligns global depth via norm matching. The full objective is:

$$\mathcal{L}(\mathbf{Y}) = \lambda_1 \cdot \mathcal{D}(S^{d_h}, S^{d_l}) + \lambda_2 \sum_i (\rho(x_i) - \rho(y_i))^2, \tag{3}$$

where $\rho(x) = \|x\|$ and $\mathcal{D}$ is typically KL divergence.

### 2.1  Problem Formulation

We consider a decentralized system with $M$ clients $\{\mathcal{C}_1, \ldots, \mathcal{C}_M\}$ coordinated by a central server owned by a private company, hospital, bank, or government agency. Each client $\mathcal{C}_m$ holds a private dataset $\mathcal{D}_m = \{\mathbf{x}_i^m\}_{i=1}^{N_m} \subset \mathbb{R}^{d_h}$, which remains local and disjoint, i.e., $\mathcal{D}_m \cap \mathcal{D}_{m'} = \emptyset$ for $m \neq m'$. Let $N = \sum_{m=1}^M N_m$ be the total number of data points, indexed globally by $i \in [N]$. We consider two real-world configurations: A) *SENSE-Pointwise*, where each client holds a single sample $\mathbf{x}^m \in \mathbb{R}^{d_h}$, and B) *SENSE-Multisite*, where each client holds a local dataset $\mathbf{X}^m = [\mathbf{x}_1^m, \ldots, \mathbf{x}_{N_m}^m] \in \mathbb{R}^{N_m \times d_h}$. Let $\mathbf{D} \in \mathbb{R}^{N \times N}$ denote the full squared distance matrix. In Euclidean space, $\mathbf{D}_{ij} = \|\mathbf{x}_i - \mathbf{x}_j\|^2$; in hyperbolic space, it reflects squared distances in the Poincaré ball $\mathbb{B}^{d_h}$ or Lorentz model $\mathbb{H}^{d_h}$ (see Appendix A.3). Due to privacy constraints, only a subset of entries is observable. Let $\Omega \subseteq [N] \times [N]$ be the set of observed indices, and define the projection operator $\mathcal{P}_\Omega : \mathbb{R}^{N \times N} \to \mathbb{R}^{N \times \overline{N}}$ as:

$$[\mathcal{P}_\Omega(\mathbf{D})]_{ij} = \begin{cases} \mathbf{D}_{ij}, & \text{if } (i,j) \in \Omega, \\ 0, & \text{otherwise.} \end{cases} \tag{4}$$

**Goal 1** *Our goal is to recover the full distance matrix* $\widehat{\mathbf{D}} \in \mathbb{R}^{N \times N}$ *from partial observations* $\mathbf{D}_\Omega = \mathcal{P}_\Omega(\mathbf{D})$ *via structured matrix completion. Instead of estimating distances directly, we infer latent embeddings* $\widehat{\mathbf{X}}$ *whose induced distances match the observed entries. This is done without access to raw features, relying solely on* $\mathbf{D}_\Omega$*. Formally,*

$$\widehat{\mathbf{D}} = \mathcal{D}(\widehat{\mathbf{X}}) = \arg\min_{\mathbf{X}'} \|\mathcal{P}_\Omega\left(\mathcal{D}(\mathbf{X}')\right) - \mathbf{D}_\Omega\|_F^2 , \tag{5}$$

*where* $\mathcal{D}(\mathbf{X}')$ *is the distance matrix induced by* $\mathbf{X}'$ *under the chosen geometry (Euclidean or hyperbolic). From* $\widehat{\mathbf{D}}$*, we derive a global low-dimensional embedding* $\mathbf{Y} = \{\mathbf{y}_i\}_{i=1}^N \subset \mathbb{R}^{d_\ell}$ *with* $d_\ell \ll d_h$*, preserving neighborhood structure.*

We use $\widehat{\mathbf{D}}$ to find the similarities, defined in Eq. 6 and optimized via divergence $\mathcal{D}(S^{d_h}, S^{d_\ell})$ (Eq. 1).

$$S_{ij}^{d_h} = \exp\left(-\frac{\widehat{\mathbf{D}}_{ij}}{2\sigma^2}\right), \quad S_{ij}^{d_\ell} = g(\|\mathbf{y}_i - \mathbf{y}_j\|^2), \tag{6}$$

For contrastive learning, we build binary similarities using $k$-nearest neighbors:

$$S_{ij}^{d_h} = \begin{cases} 1, & \text{if } j \in \text{kNN}(i; \widehat{\mathbf{D}}), \\ 0, & \text{otherwise,} \end{cases} \quad S_{ij}^{d_\ell} = \phi(\mathbf{y}_i, \mathbf{y}_j) = \frac{1}{1 + \|\mathbf{y}_i - \mathbf{y}_j\|^2}, \tag{7}$$

and minimize the contrastive loss (Eq. 2). For hierarchical data, we apply CO-SNE, treating $\widehat{\mathbf{D}}$ as squared hyperbolic distances in the Poincaré model to compute similarities (Eq. 17 in Appendix). The embedding $\mathbf{Y} \subset \mathbb{B}^{d_\ell}$ is optimized using the CO-SNE loss (Eq. 3).

**Remark 1** *Conventional FL methods (e.g., FedAvg) assume large local datasets, require multiple communication rounds, and expose gradients that risk privacy leaks [20, 62]. They also fail in pointwise settings where local training is infeasible. In contrast, SENSE reconstructs* $\widehat{\mathbf{D}}$ *via privacy-preserving matrix completion and then optimizes NE, CNE, or CO-SNE objectives without sharing raw features.*

## 3 Proposed Framework: SENSE

As described in Section 2.1, we consider two decentralized settings: *SENSE-Pointwise* and *SENSE-Multisite*. In both, each client holds private non-anchor (NA) data and accesses a shared anchor set $\mathcal{A} = \{a_1, \ldots, a_K\}$ with feature matrix $\mathbf{X}_A = [\mathbf{p}_1, \ldots, \mathbf{p}_K]^\top \in \mathbb{R}^{K \times d_h}$. Anchors, broadcast by the server, may be global or client-specific (see Appendix A.8). Let $\mathcal{X} = \{x_1, \ldots, x_N\}$ be the set of all private NA points, where $N = \sum_{m=1}^M N_m$. Each client computes squared distances between its NAs and accessible anchors:

$$\mathbf{d}_i^m = \left[\|x_i^m - \mathbf{p}_1\|^2, \ldots, \|x_i^m - \mathbf{p}_K\|^2\right],$$

and transmits these to the server, masking unshared local anchors. In *Pointwise*, each client contributes one NA-anchor vector, in *Multisite*, intra-client NA–NA distances may also be known. The global incomplete squared distance matrix $\mathbf{D} \in \mathbb{R}^{(K+N) \times (K+N)}$ is partitioned as:

$$\mathbf{D} = \begin{bmatrix} E & F \\ F^\top & G \end{bmatrix}, \tag{8}$$

where $E$ is anchor–anchor, $F$ is anchor–NA, and $G$ is NA–NA. The observed subset is indexed by $\Omega \subseteq [K + N]^2$, based on anchor visibility and client configuration. We consider four configurations: *Pointwise-Full*, *Pointwise-Partial*, *Multisite-Full*, and *Multisite-Partial* which differ in the extent of observed entries in $F$ (anchor–NA) and $G$ (NA–NA). These define distinct visibility patterns in $\Omega$, summarized in Appendix Table 4 and illustrated in Figure 1, and determine which distances are available for structured matrix completion.

To reconstruct the full matrix $\widehat{\mathbf{D}}$, or specifically $\widehat{G}$, we apply geometry-specific solvers: anchored-MDS in Euclidean space (discussed in Sec 3.1) and LHydra [30] in hyperbolic space. The complete pipeline is outlined in Algorithm 1 in Appendix.

**Remark 2** *In practice, $F$ may be only partially visible due to bandwidth, privacy, or data limitations. SENSE is designed to operate under such conditions. Whether $F$ is full or partial, structured matrix completion (in SENSE) enables accurate and privacy-preserving recovery of inter-client affinities.*

### 3.1 SENSE via Anchored-MDS

Classical MDS embeds $N$ points by minimizing stress over a fully observed distance matrix $\mathbf{D} \in \mathbb{R}^{N \times N}$. The embedding $\mathbf{X} \in \mathbb{R}^{N \times d_h}$ minimizes:

$$\sigma(\mathbf{X}) = \sum_{i<j} \left( \|x_i - x_j\| - \delta_{ij} \right)^2,$$

where $\delta_{ij}$ is the input Euclidean distance between points $i$ and $j$. SMACOF solves this using a majorization-based surrogate [13], $\tau(\mathbf{X}, \mathbf{Z}) = C + \operatorname{tr}(\mathbf{X}^\top \mathbf{V} \mathbf{X}) - 2 \operatorname{tr}(\mathbf{X}^\top \mathbf{B}(\mathbf{Z})\mathbf{Z})$, with the iterative update:

$$\mathbf{X}^{(k)} = \mathbf{V}^\dagger \mathbf{B}(\mathbf{X}^{(k-1)})\mathbf{X}^{(k-1)}. \tag{9}$$

In SENSE, the full distance matrix $\mathbf{D}$ is not available, instead we work with a structured, incomplete matrix of observed anchor–NA distances. Let the embedding be $\mathbf{X} = [\mathbf{X}_A\ \mathbf{X}_{NA}]^\top$, where $\mathbf{X}_A$ and $\mathbf{X}_{NA}$ are anchor and NA embeddings, respectively. The stress is minimized over observed entries only:

$$\sigma(\mathbf{X}) = \|\mathcal{P}_\Omega(\mathcal{D}(\mathbf{X}) - \mathbf{D})\|_F^2,$$

where $\mathcal{P}_\Omega$ projects onto the observed indices $\Omega$, and $\mathcal{D}(\mathbf{X})$ computes pairwise distances. The SMACOF updates are restricted to $\Omega$, with:

$$V_{ij} = \begin{cases} |\{j : (i,j) \in \Omega\}|, & i = j \\ -1, & (i,j) \in \Omega,\ i \neq j \\ 0, & \text{otherwise} \end{cases}, \quad B_{ij}(\mathbf{X}) = \begin{cases} -\frac{\delta_{ij}}{\|x_i - x_j\|}, & (i,j) \in \Omega,\ i \neq j \\ -\sum_{k \neq i,\ (i,k) \in \Omega} B_{ik}, & i = j \\ 0, & \text{otherwise} \end{cases}$$

We partition $V$ and $B$ as defined in Eq. 10, where $V_{AA}, B_{AA} \in \mathbb{R}^{K \times K}$, $V_{AN}, B_{AN} \in \mathbb{R}^{K \times N}$, and $V_{NN}, B_{NN} \in \mathbb{R}^{N \times N}$:

$$\mathbf{V} = \begin{bmatrix} \mathbf{V}_{AA} & \mathbf{V}_{AN} \\ \mathbf{V}_{AN}^\top & \mathbf{V}_{NN} \end{bmatrix}, \quad \mathbf{B} = \begin{bmatrix} \mathbf{B}_{AA} & \mathbf{B}_{AN} \\ \mathbf{B}_{AN}^\top & \mathbf{B}_{NN} \end{bmatrix} \tag{10}$$

The update rule for NA embeddings becomes:

$$\mathbf{X}_{NA}^{(k)} = \mathbf{V}_{NN}^\dagger \left( \mathbf{B}_{NN}\mathbf{X}_{NA}^{(k-1)} + \mathbf{B}_{AN}^\top \mathcal{P}_\Omega(\mathbf{X}_A) - \mathbf{V}_{AN}^\top \mathcal{P}_\Omega(\mathbf{X}_A) \right). \tag{11}$$

This projection-aware update ensures $\mathbf{X}_{NA}$ uses only observed/available distances, enabling privacy-preserving global embedding under any SENSE configuration. The projection operator $\mathcal{P}_\Omega$ acts as a binary mask over observed entries. While $\mathbf{V}$ and $\mathbf{B}$ are derived from $\Omega$, we apply $\mathcal{P}_\Omega$ to $\mathbf{X}_A$ in Eq. (11) to retain only anchors with observed anchor–NA distances. This avoids leakage from inaccessible anchors and ensures privacy-compliant updates. Pseudocode is provided in Appendix A.7. Furthermore, to preserve privacy, the number of shared anchors $K$ must be limited. Theorems 3.1, 3.2 (Euclidean) and Lemma 1 (hyperbolic) characterize how $K$ relates to embedding dimension $d_h$ across SENSE configurations, establishing conditions for faithful reconstruction.

**Theorem 3.1** *Let $\mathcal{X} = \{\mathbf{x}_1, \ldots, \mathbf{x}_N\} \subset \mathbb{R}^{d_h}$ be the set of NA data points, and let $\mathcal{A} = \{\mathbf{a}_1, \ldots, \mathbf{a}_K\} \subset \mathbb{R}^{d_h}$ be the set of $K$ anchor points. Suppose we observe the pairwise Euclidean distances $\{\|\mathbf{x}_i - \mathbf{a}_j\|\}_{i \in [N], j \in [K]}$ between each NA and all anchors. If the number of anchors satisfies $K < d_h$, then the original NA features $\{\mathbf{x}_i\}_{i=1}^N$ cannot be exactly reconstructed from these distances, guaranteeing the privacy of the individual client data.*

*Proof.* Deferred in Appendix, check A.2.

SENSE supports multiple configurations, which critically influence embedding fidelity and privacy. Theorem 3.2 formalizes privacy guarantees when only partial anchor–NA distances (block $F$) are available, covering both *pointwise* and *multisite* regimes. *1) SENSE-Pointwise:* Each client $j \in [N]$ holds a single private point $\boldsymbol{x}_j \in \mathbb{R}^{d_h}$ and accesses a subset of anchors indexed by $\mathcal{I}_j \subseteq [K]$. The corresponding anchor set is $\mathcal{A}_j = \{\boldsymbol{a}_i\}_{i \in \mathcal{I}_j}$, comprising: (i) global anchors $\mathcal{A}_G = \{\boldsymbol{a}_1, \ldots, \boldsymbol{a}_{M_G}\}$, shared across all clients, and (ii) local anchors $\mathcal{A}_L^{(j)}$, unique to client $j$. The total number of anchors observed is $r_j = |\mathcal{I}_j| = M_G + M_L^{(j)}$. *2) SENSE-Multisite:* Each client $m \in [M]$ holds a local dataset

$\mathcal{X}^{(m)} = \{\boldsymbol{x}_{m,1}, \ldots, \boldsymbol{x}_{m,n_m}\} \subset \mathbb{R}^{d_h}$, where $N = \sum_{m=1}^{M} n_m$. Each point $\boldsymbol{x}_{m,i}$ observes distances to (i) a shared global anchor set $\mathcal{A}_G$, and (ii) a local anchor set $\mathcal{A}_L^{(m)}$ exclusive to client $m$. Let $\mathcal{I}_{m,i} = \mathcal{I}_G \cup \mathcal{I}_L^{(m)}$ be the index set of accessible anchors, with $r_{m,i} = |\mathcal{I}_{m,i}|$ denoting the number observed.

**Theorem 3.2** *Let $\mathcal{X} = \{\boldsymbol{x}_1, \ldots, \boldsymbol{x}_N\} \subset \mathbb{R}^{d_h}$ be the set of all non-anchor (NA) points across all clients, where each $\boldsymbol{x}_i$ computes squared distances only to a subset of accessible anchors $\mathcal{A}_i = \{\boldsymbol{a}_j\}_{j \in \mathcal{I}_i}$, with $|\mathcal{I}_i| = r_i$. If $r_i < d_h$ for all $i \in [N]$, then exact recovery of each $\boldsymbol{x}_i$ is impossible. The inverse map from anchor distances to features is non-unique, preserving privacy under both pointwise and multisite configurations.*

*Proof.* Defered in Appendix, check A.1.

**Lemma 1** *Let $\{x_1, \ldots, x_{K+N}\} \subset \mathbb{H}^{d_h}$ be $K$ anchors and $N$ non-anchor points in hyperbolic space with curvature $-\kappa$. Suppose only blocks $E$ and $F$ of the global distance matrix $\mathbf{D}$ are observed. If $K < d_h$, the NA coordinates cannot be exactly recovered up to isometry in $\mathbb{H}^{d_h}$, ensuring the privacy of the client data in SENSE. This follows from the contrapositive of the L-HYDRA theorem [30], which guarantees exact recovery only when $K \geq d_h$ and anchors span a full subspace.*

## 3.2 SENSE in Evolving Distributed Environments

In dynamic settings, new data points arrive continuously e.g., a hospital admitting a patient, a bank processing a transaction, or a platform onboarding a user. Recomputing the full embedding for each arrival is inefficient and may disrupt global structure. Existing decentralized NE methods [33, 43, 47, 48] assume static datasets and lack support for incremental updates, making them unsuitable for streaming environments. SENSE, by contrast, is modular and compatible with out-of-sample embedding methods [5, 24, 41]. Once the global embedding is constructed via anchor-based completion and NE optimization, it defines a geometry-aware coordinate space that supports new points without full recomputation. Let $\mathbf{X}_{NA} = [\mathbf{x}_1, \ldots, \mathbf{x}_N] \in \mathbb{R}^{N \times d_h}$ be the reconstructed NA embeddings. When a new point $\mathbf{y}$ arrives, we select $K$ existing points as pseudo-anchors $\mathcal{A} = \{a_1, \ldots, a_K\} \subset \mathbf{X}_{NA}$, with coordinates $\mathbf{X}_A = [\mathbf{p}_1, \ldots, \mathbf{p}_K]^\top \in \mathbb{R}^{K \times d_h}$. Given dissimilarities $\{\delta_{l_i y}\}_{i=1}^{K}$ to these anchors, we compute the embedding $\hat{\mathbf{y}}$ by solving:

$$\hat{\sigma}(\hat{\mathbf{y}}) = \sum_{i=1}^{K} \left( \|\mathbf{p}_i - \hat{\mathbf{y}}\|_2 - \delta_{l_i y} \right)^2. \tag{12}$$

Here, $\delta_{l_i y}$ is the dissimilarity in the original space, and $\|\mathbf{p}_i - \hat{\mathbf{y}}\|_2$ is the distance in the embedding space. Only $\hat{\mathbf{y}}$ is optimized, anchors remain fixed. Since $K < d_h$, exact recovery is impossible (Theorems 3.1, 3.2), ensuring privacy. This lightweight optimization requires no raw data and supports real-time integration, making SENSE well-suited for scalable, privacy-constrained systems.

# 4 Experiments

In this section, we first outline the experimental setup, followed by an evaluation of SENSE across diverse datasets and deployment settings.

## 4.1 Experimental Setup

**Datasets.** We evaluate SENSE on 14 public datasets widely used in DR and representation learning [18, 63]. These include three benchmarks: MNIST [14], Fashion-MNIST [58], and CIFAR-10 [21]; seven MedMNIST datasets [60]: DermaMNIST, PneumoniaMNIST, RetinaMNIST, BreastMNIST, BloodMNIST, OrganCMNIST, OrganSMNIST; and the German Credit dataset [25] for financial risk modeling. For hyperbolic evaluation, we use three graph datasets: Airport [36], Amazon [59], and DBLP [29]. Detailed dataset statistics and system specifications are provided in Appendix Table 5 and A.12.

**Baselines.** We compare SENSE against centralized (Van) baselines: t-SNE [53], UMAP [37], PHATE [38], and CNE [11] (with $s \in \{0, 0.5, 1\}$). These assume full raw data access at a central server and serve as upper bounds for evaluating SENSE's privacy-preserving performance.

**Implementation Details.** SENSE comprises two stages: matrix completion and global embedding. In the first stage, data is partitioned across $M$ clients. In *Pointwise*, each client holds one NA point, sampled randomly. In *Multisite*, clients hold multiple NA points under IID or non-IID splits

Table 1: Full vs. Partial comparison in MULTISITE under non-IID (unbalanced) splits. Evaluation spans centralized and privacy-preserving SENSE variants across different embedding quality metrics.

| Data | Metric | t-SNE | | UMAP | | PHATE | | CNE(s=0) | | CNE(s=0.5) | | CNE(s=1) | |
|---|---|---|---|---|---|---|---|---|---|---|---|---|---|
| | | VAN. | SENSE | VAN. | SENSE | VAN. | SENSE | VAN. | SENSE | VAN. | SENSE | VAN. | SENSE |
| — Multisite-Partial Setting — | | | | | | | | | | | | | |
| MNIST | Trust. | 0.9890 | 0.9898 | 0.9553 | 0.9552 | 0.8741 | 0.8763 | 0.9517 | 0.9521 | 0.9524 | 0.9538 | 0.9455 | 0.9476 |
| | Cont. | 0.9575 | 0.9639 | 0.9774 | 0.9771 | 0.9811 | 0.9804 | 0.9806 | 0.9797 | 0.9799 | 0.9787 | 0.9799 | 0.9787 |
| | Stead. | 0.7719 | 0.7861 | 0.7639 | 0.7635 | 0.6628 | 0.6746 | 0.7840 | 0.7790 | 0.7752 | 0.7768 | 0.7634 | 0.7658 |
| | Cohes. | 0.8189 | 0.8458 | 0.8865 | 0.8853 | 0.8668 | 0.8877 | 0.9229 | 0.9112 | 0.9107 | 0.9196 | 0.9158 | 0.9087 |
| fashionMNIST | Trust. | 0.9902 | 0.9914 | 0.9140 | 0.9148 | 0.9579 | 0.9557 | 0.9765 | 0.9752 | 0.9784 | 0.9769 | 0.9765 | 0.9731 |
| | Cont. | 0.9608 | 0.9590 | 0.9812 | 0.9818 | 0.9910 | 0.9906 | 0.9915 | 0.9913 | 0.9905 | 0.9903 | 0.9900 | 0.9901 |
| | Stead. | 0.8415 | 0.8643 | 0.7570 | 0.7622 | 0.7836 | 0.7891 | 0.8632 | 0.8638 | 0.8643 | 0.8660 | 0.8493 | 0.8513 |
| | Cohes. | 0.6496 | 0.6559 | 0.6748 | 0.7069 | 0.7051 | 0.7115 | 0.7680 | 0.7669 | 0.7637 | 0.7508 | 0.7792 | 0.7666 |
| — Multisite-Full Setting — | | | | | | | | | | | | | |
| MNIST | Trust. | 0.9890 | 0.9852 | 0.9553 | 0.9570 | 0.8741 | 0.8780 | 0.9517 | 0.9516 | 0.9524 | 0.9542 | 0.9455 | 0.9452 |
| | Cont. | 0.9575 | 0.9518 | 0.9774 | 0.9754 | 0.9811 | 0.9797 | 0.9806 | 0.9772 | 0.9799 | 0.9763 | 0.9799 | 0.9761 |
| | Stead. | 0.7719 | 0.7953 | 0.7639 | 0.7726 | 0.6628 | 0.6688 | 0.7840 | 0.7808 | 0.7752 | 0.7828 | 0.7634 | 0.7690 |
| | Cohes. | 0.8189 | 0.8328 | 0.8865 | 0.8665 | 0.8668 | 0.8818 | 0.9229 | 0.9047 | 0.9107 | 0.8926 | 0.9158 | 0.9106 |
| fashionMNIST | Trust. | 0.9902 | 0.9895 | 0.9140 | 0.9076 | 0.9579 | 0.9555 | 0.9765 | 0.9752 | 0.9784 | 0.9769 | 0.9765 | 0.9725 |
| | Cont. | 0.9608 | 0.9731 | 0.9812 | 0.9797 | 0.9910 | 0.9902 | 0.9915 | 0.9906 | 0.9905 | 0.9895 | 0.9900 | 0.9891 |
| | Stead. | 0.8415 | 0.8604 | 0.7570 | 0.7530 | 0.7836 | 0.7981 | 0.8632 | 0.8608 | 0.8643 | 0.8649 | 0.8493 | 0.8538 |
| | Cohes. | 0.6496 | 0.6936 | 0.6748 | 0.7019 | 0.7051 | 0.7039 | 0.7680 | 0.7503 | 0.7637 | 0.7591 | 0.7792 | 0.7695 |
| — Pointwise-Full Setting — | | | | | | | | | | | | | |
| MNIST | Trust. | 0.9661 | 0.9679 | 0.9484 | 0.9467 | 0.8457 | 0.8469 | 0.9218 | 0.9166 | 0.9164 | 0.9138 | 0.9137 | 0.9151 |
| | Cont. | 0.9418 | 0.9410 | 0.9376 | 0.9396 | 0.9546 | 0.9538 | 0.9434 | 0.9422 | 0.9428 | 0.9417 | 0.9409 | 0.9403 |
| | Stead. | 0.8083 | 0.8113 | 0.7878 | 0.7763 | 0.6953 | 0.6958 | 0.8024 | 0.8003 | 0.8041 | 0.7996 | 0.8025 | 0.7914 |
| | Cohes. | 0.7904 | 0.7998 | 0.7855 | 0.7819 | 0.7912 | 0.7843 | 0.7988 | 0.7982 | 0.8034 | 0.7894 | 0.7931 | 0.7919 |
| fashionMNIST | Trust. | 0.9647 | 0.9681 | 0.9441 | 0.9434 | 0.8407 | 0.8375 | 0.9283 | 0.9264 | 0.9255 | 0.9245 | 0.9256 | 0.9196 |
| | Cont. | 0.9430 | 0.9454 | 0.9386 | 0.9373 | 0.9542 | 0.9528 | 0.9464 | 0.9460 | 0.9456 | 0.9440 | 0.9451 | 0.9429 |
| | Stead. | 0.8118 | 0.8103 | 0.7797 | 0.7779 | 0.6923 | 0.6931 | 0.8087 | 0.8049 | 0.8085 | 0.8003 | 0.8082 | 0.8150 |
| | Cohes. | 0.7570 | 0.7882 | 0.7685 | 0.7670 | 0.7564 | 0.7599 | 0.7876 | 0.7786 | 0.7843 | 0.7788 | 0.7838 | 0.7710 |

(balanced/unbalanced). A subset of $10\%$ of the total data points is designated as anchors. In *Full* settings, all anchors are global, and in *Partial*, anchors are split into global and client-specific local sets. The total number of anchors (global + local) is fixed at $d_h - 1$, where $d_h$ is the original feature dimension. In the embedding stage, we use the completed global distance matrix to generate privacy-preserving embeddings using multiple neighbor embedding methods. For Euclidean geometry, we use the official implementations of t-SNE [53], UMAP [37], and PHATE (via its standard Python library). For CNE, we adopt the implementation from [11], where the parameter $s$ controls the attraction-repulsion tradeoff: $s = 0$ mimics t-SNE, $s = 1$ aligns with UMAP, and intermediate values interpolate between them. CNE operates within a contrastive learning framework using negative sampling. For hyperbolic embeddings, we use the CO-SNE implementation from [22].

**Data Partitioning.** To simulate realistic distributed settings, we evaluate SENSE under both IID and non-IID distributions using Dirichlet-based partitioning. For each class $c$, client-wise proportions are drawn from $q_c \sim \text{Dir}(\alpha)$, where lower $\alpha$ yields greater heterogeneity and class imbalance [55, 61]. We set $\alpha = 0.5$ in all experiments. Three partitioning schemes are used: *IID* (uniform class mix), *non-IID balanced* (varying class distributions, equal client sizes), and *non-IID unbalanced* (both class and size vary).

**Evaluation Metrics.** We assess SENSE using both reconstruction and embedding quality metrics. For fidelity, we compute *Relative Distance Error (DE)* and *F-score (FS)* between the reconstructed distance matrix (NA-NA) $\hat{G}$ and ground truth $G_{\text{true}}$: $\text{DE} = \frac{\|\hat{G} - G_{\text{true}}\|_F}{\|G_{\text{true}}\|_F}$, and $\text{FS} = \frac{2\,\text{tp}}{2\,\text{tp}+\text{fp}+\text{fn}}$, where tp, fp, and fn are true, false positive, and false negative neighbors respectively [17]. To evaluate 2D embeddings, we compute *Trustworthiness* and *Continuity* [54], which measure neighborhood agreement between original and embedded spaces. We also report *Steadiness* and *Cohesiveness* [26] to assess global structural reliability: steadiness detects false groupings and cohesiveness quantifies how well true input clusters are preserved.

**4.2 Result Analysis.**
We comprehensively evaluate SENSE across: *1) Standard image datasets (MNIST, FashionMNIST, CIFAR-10):* These are evaluated under *Pointwise-Full*, *Multisite-Full*, and *Multisite-Partial* with non-IID unbalanced splits. As shown in Table 1 and in Appendix 8, SENSE closely matches centralized baselines across Cont., Trust., Stead., and Cohes. Notably, the *Partial* configuration performs comparably to *Full*, indicating that accurate reconstruction of the global distance matrix is possible even with partial anchor–NA observations. Table 7 further confirms high F-score and low distance error, validating strong neighborhood preservation under strict privacy constraints.
*2) MedMNIST datasets:* These are evaluated across unbalanced non-IID, balanced non-IID, and IID splits. SENSE consistently matches centralized performance (Tables 2,10,9), even under high

Table 2: Performance of centralized (Van.) and SENSE variants under non-IID unbalanced splits.

| Data | Metric | t-SNE | | UMAP | | PHATE | | CNE(s=0) | | CNE(s=0.5) | | CNE(s=1) | |
|---|---|---|---|---|---|---|---|---|---|---|---|---|---|
| | | VAN. | SENSE | VAN. | SENSE | VAN. | SENSE | VAN. | SENSE | VAN. | SENSE | VAN. | SENSE |
| PneumoniaMNIST | Trust. | 0.9723 | 0.9712 | 0.7699 | 0.7673 | 0.8570 | 0.8590 | 0.9027 | 0.9008 | 0.8976 | 0.8952 | 0.8832 | 0.8806 |
| | Cont. | 0.9418 | 0.9383 | 0.9140 | 0.9154 | 0.9624 | 0.9608 | 0.9594 | 0.9591 | 0.9590 | 0.9583 | 0.9606 | 0.9599 |
| | Stead. | 0.7868 | 0.7932 | 0.6258 | 0.6168 | 0.7247 | 0.7204 | 0.7552 | 0.7591 | 0.7496 | 0.7461 | 0.7283 | 0.7341 |
| | Cohes. | 0.6991 | 0.6591 | 0.6318 | 0.6250 | 0.6953 | 0.6957 | 0.6983 | 0.7085 | 0.7052 | 0.7142 | 0.7015 | 0.7065 |
| BloodMNIST | Trust. | 0.9633 | 0.9609 | 0.8674 | 0.8632 | 0.8493 | 0.8513 | 0.8841 | 0.8816 | 0.8814 | 0.8795 | 0.8737 | 0.8715 |
| | Cont. | 0.9256 | 0.9375 | 0.9411 | 0.9401 | 0.9435 | 0.9428 | 0.9555 | 0.9552 | 0.9558 | 0.9556 | 0.9555 | 0.9552 |
| | Stead. | 0.7498 | 0.7480 | 0.6889 | 0.6874 | 0.6781 | 0.6851 | 0.7172 | 0.7323 | 0.7186 | 0.7216 | 0.7100 | 0.7132 |
| | Cohes. | 0.7242 | 0.7178 | 0.7253 | 0.7253 | 0.7456 | 0.7448 | 0.7462 | 0.7440 | 0.7384 | 0.7540 | 0.7533 | 0.7379 |
| BreastMNIST | Trust. | 0.9379 | 0.9378 | 0.7817 | 0.7998 | 0.8921 | 0.8884 | 0.9133 | 0.9117 | 0.9124 | 0.9113 | 0.9108 | 0.9108 |
| | Cont. | 0.9508 | 0.9481 | 0.8140 | 0.8247 | 0.9616 | 0.9563 | 0.9519 | 0.9515 | 0.9516 | 0.9513 | 0.9510 | 0.9509 |
| | Stead. | 0.8417 | 0.8329 | 0.5605 | 0.5550 | 0.8037 | 0.8149 | 0.8438 | 0.8480 | 0.8491 | 0.8495 | 0.8490 | 0.8398 |
| | Cohes. | 0.6091 | 0.6137 | 0.4095 | 0.4112 | 0.5668 | 0.5570 | 0.5777 | 0.5695 | 0.5807 | 0.5689 | 0.5675 | 0.5585 |
| DermaMNIST | Trust. | 0.9757 | 0.9770 | 0.7496 | 0.7466 | 0.8737 | 0.8728 | 0.9130 | 0.9121 | 0.9119 | 0.9116 | 0.9020 | 0.9021 |
| | Cont. | 0.9461 | 0.9572 | 0.9127 | 0.9122 | 0.9736 | 0.9730 | 0.9709 | 0.9713 | 0.9706 | 0.9707 | 0.9716 | 0.9715 |
| | Stead. | 0.7977 | 0.7979 | 0.5945 | 0.5936 | 0.7308 | 0.7319 | 0.7739 | 0.7689 | 0.7682 | 0.7686 | 0.7578 | 0.7553 |
| | Cohes. | 0.7147 | 0.7111 | 0.5586 | 0.5459 | 0.7127 | 0.7108 | 0.7268 | 0.7321 | 0.7385 | 0.7502 | 0.7438 | 0.7383 |
| RetinaMNIST | Trust. | 0.9797 | 0.9736 | 0.8793 | 0.8636 | 0.9161 | 0.9050 | 0.9486 | 0.9357 | 0.9475 | 0.9348 | 0.9451 | 0.9336 |
| | Cont. | 0.9496 | 0.9669 | 0.9273 | 0.9244 | 0.9738 | 0.9734 | 0.9720 | 0.9714 | 0.9707 | 0.9701 | 0.9678 | 0.9680 |
| | Stead. | 0.8442 | 0.8498 | 0.6307 | 0.5923 | 0.7559 | 0.7636 | 0.8267 | 0.8176 | 0.8196 | 0.8138 | 0.8158 | 0.8040 |
| | Cohes. | 0.6734 | 0.7281 | 0.5832 | 0.5828 | 0.6957 | 0.6991 | 0.7100 | 0.7137 | 0.7089 | 0.6982 | 0.6883 | 0.6990 |
| OrganCMNIST | Trust. | 0.9621 | 0.9387 | 0.8887 | 0.8867 | 0.8850 | 0.8871 | 0.9134 | 0.9041 | 0.9159 | 0.9056 | 0.9019 | 0.8907 |
| | Cont. | 0.9207 | 0.9170 | 0.9268 | 0.9247 | 0.9691 | 0.9699 | 0.9733 | 0.9693 | 0.9729 | 0.9685 | 0.9737 | 0.9696 |
| | Stead. | 0.7011 | 0.7855 | 0.7527 | 0.7718 | 0.7935 | 0.8093 | 0.8666 | 0.8755 | 0.8733 | 0.8722 | 0.8597 | 0.8607 |
| | Cohes. | 0.4685 | 0.5037 | 0.3322 | 0.3373 | 0.5431 | 0.5444 | 0.4653 | 0.5096 | 0.5681 | 0.5233 | 0.5745 | 0.5375 |
| OrganSMNIST | Trust. | 0.9552 | 0.9357 | 0.8741 | 0.8625 | 0.8792 | 0.8821 | 0.9114 | 0.9028 | 0.9126 | 0.9040 | 0.8993 | 0.8912 |
| | Cont. | 0.9214 | 0.9169 | 0.9246 | 0.9213 | 0.9684 | 0.9700 | 0.9738 | 0.9682 | 0.9731 | 0.9675 | 0.9736 | 0.9683 |
| | Stead. | 0.6765 | 0.7311 | 0.7222 | 0.7485 | 0.7809 | 0.7995 | 0.8609 | 0.8659 | 0.8664 | 0.8708 | 0.8561 | 0.8582 |
| | Cohes. | 0.4951 | 0.4814 | 0.3603 | 0.3211 | 0.5198 | 0.5343 | 0.4704 | 0.44009 | 0.5192 | 0.4833 | 0.5155 | 0.5033 |
| german-credit | Trust. | 0.9745 | 0.9543 | 0.9514 | 0.9294 | 0.8555 | 0.8394 | 0.9337 | 0.9124 | 0.9380 | 0.9072 | 0.9336 | 0.9092 |
| | Cont. | 0.9583 | 0.9424 | 0.9604 | 0.9410 | 0.9481 | 0.9255 | 0.9571 | 0.9438 | 0.9576 | 0.9438 | 0.9571 | 0.9440 |
| | Stead. | 0.8576 | 0.8248 | 0.8313 | 0.7933 | 0.7483 | 0.7061 | 0.8398 | 0.7921 | 0.8479 | 0.7855 | 0.8436 | 0.7906 |
| | Cohes. | 0.6774 | 0.6755 | 0.6638 | 0.6568 | 0.6893 | 0.6745 | 0.6446 | 0.6551 | 0.6575 | 0.6481 | 0.6513 | 0.6676 |

heterogeneity. Table 6 in Appendix, further shows low DE and high FS, confirming strong structural and similarity preservation.

*3) Hyperbolic datasets (Airport, Amazon, DBLP):* For these datasets, the results in Table 3 highlight SENSE's geometry-aware design, achieving high FS and very low DE in non-Euclidean spaces. This confirms its adaptability across geometric regimes. Overall, SENSE effectively ensures:

- *Neighbor preservation:* High continuity and trustworthiness show SENSE keeps similar points close in the embedding, preserving semantics across clients.
- *Similarity recovery:* Despite no raw data access, SENSE accurately approximates pairwise distances evidenced by low DE and high FS.
- *Cluster structure:* Comparable steadiness and cohesiveness confirm that SENSE maintains cluster alignment without fragmentation.

**Visualization.** Figure 2 shows global embeddings learned by SENSE on MNIST in the MULTISITE setting with 25,000 non-anchor samples across 10 clients in an unbalanced non-IID split. Using only 783 anchors ($d_h - 1$), SENSE constructs high-quality embeddings without accessing or sharing raw features. Embeddings from t-SNE, UMAP, PHATE, and CNE cleanly separate semantic groups, preserving local neighborhoods and global cluster topology. By estimating inter-client similarities, SENSE enables meaningful inter-client positive/negative contrastive pairs. This highlights its ability to learn structure-preserving, privacy-compliant embeddings in decentralized, heterogeneous settings. Additional visualizations are in the Appendix.

Table 3: FS and DE for hyperbolic datasets in POINTWISE setting.

| Dataset | FS | DE |
|---|---|---|
| AIRPORT | 0.9992 | 0.000067 |
| AMAZON | 0.9945 | 0.00052 |
| DBLP | 0.9929 | 0.00073 |

### 4.3 Ablation Study.

To validate Theorems 3.1, 3.2, and Lemma 1, we perform an ablation study by varying anchor count from $d_h - \epsilon$ to $d_h + \epsilon$. We evaluate SENSE using five normalized metrics, plotted in Figure 3: (i) *Cosine Similarity* [39] between ground-truth $X'_{\mathrm{NA}}$ and reconstructed latent embeddings $\widehat{X}_{\mathrm{NA}}$; (ii) *Distance Error* and (iii) *F-score* (Sec. 4.1); (iv) *Pearson Correlation* ($\rho$) [49] over NA–NA distances; and (v) *Frobenius Norm Error* ($X_{\mathrm{frob}}$) [28], capturing reconstruction loss (full definitions in Appendix A.14). Key observations from the study:

- *Effective with few anchors:* Even with anchor count well below $d_h$ (e.g., $d_h - 100$), SENSE achieves high F-score, low distance error, and strong cosine similarity, showing robust neighborhood preservation in resource-constrained settings.

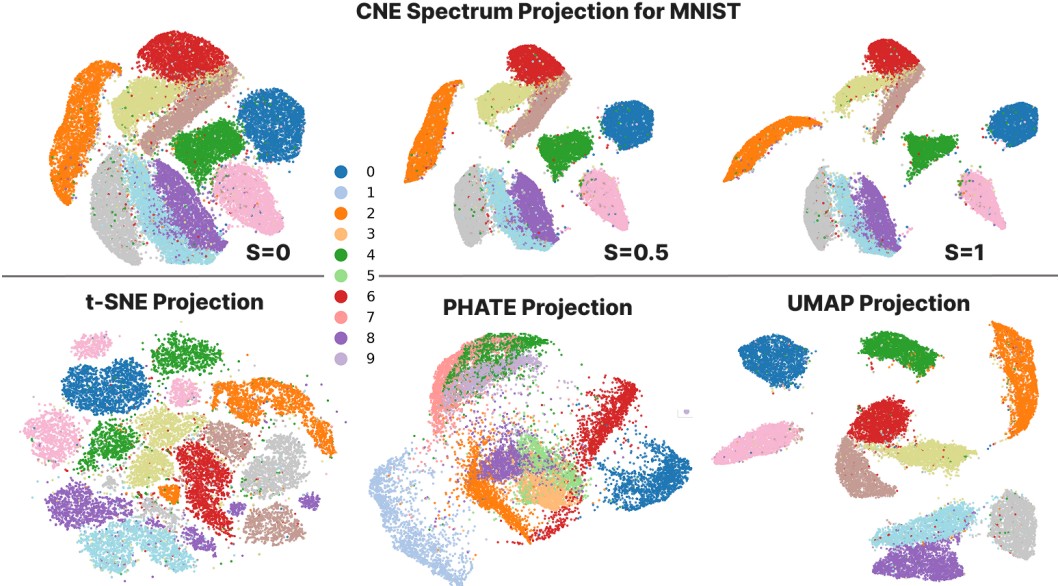

Figure 2: Global embeddings of MNIST under the MULTISITE setting. **Top:** CNE spectrum with SENSE. **Bottom:** t-SNE, PHATE, and UMAP embeddings generated via SENSE without any raw feature sharing. All embeddings preserve global structure while ensuring privacy.

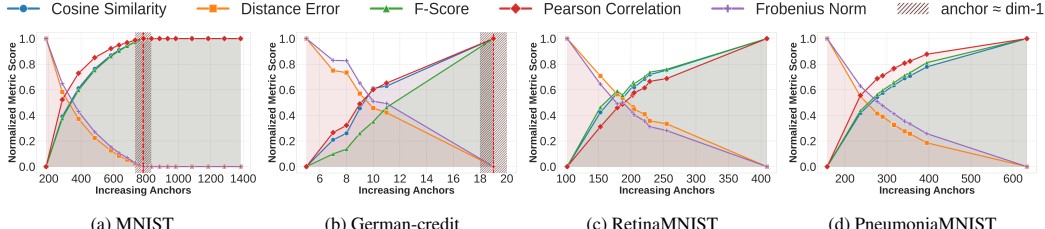

Figure 3: Impact of anchor count on normalized metric scores under non-IID unbalanced distributions. The red vertical line denotes the theoretical privacy threshold at $d_h - 1$ (783 for MNIST, 19 for German Credit), beyond which exact recovery may be possible. For Retina and Pneumonia, this threshold lies outside the x-axis range, resulting in monotonic performance gains. Trends confirm trade-offs between reconstruction fidelity and privacy risk as anchor count increases.

- *Privacy-compliant reconstruction:* As anchors approach $d_h$, cosine and Pearson scores improve. Beyond $d_h + 1$, near-zero Frobenius error indicates possible exact recovery highlighting the need to limit anchor count to preserve privacy.
- *Structural consistency:* Pearson correlation rises with anchor count, saturating near 1.0 at $d_h + 1$, with corresponding drops in Frobenius error confirming theoretical bounds for exact recovery.
- *Metric alignment with theoretical thresholds:* Across datasets, all metrics converge near $d_h$, with diminishing gains beyond matching theoretical thresholds.

These results validate that SENSE achieves high-fidelity, privacy-compliant reconstruction with minimal anchors, making it scalable and effective in decentralized settings with limited observability.

## 5 Conclusion

We propose SENSE, a unified geometry-aware framework for decentralized neighbor embedding that enables global projections without raw data exchange. SENSE addresses the key challenge of missing inter-client similarities via structured matrix completion using anchor-based distance observations. It supports both Euclidean and hyperbolic spaces and adapts to four practical deployment settings. By reconstructing global distance geometry from sparse, client-local views, SENSE accurately approximates both attractive–repulsive (NE) and positive–negative (CNE) interactions, while limiting anchor count to preserve privacy. The completed matrix enables classical and contrastive neighbor embeddings under strong privacy guarantees. Extensive experiments show that SENSE closely matches centralized baselines in neighborhood and cluster preservation across diverse non-IID scenarios. Theoretical results provide conditions for both faithful reconstruction and formal privacy protection, making SENSE a scalable and secure solution for distributed representation learning.

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

# A Appendix

## A.1 Neighbor Embedding (NE).

**Definition A.1** *t-SNE models $p_{ij}$ as symmetrized conditional probabilities using Gaussian kernels:* $p_{j|i} \propto \exp(-\|x_i - x_j\|^2/2\sigma_i^2)$, *with* $p_{ij} = \frac{p_{j|i}+p_{i|j}}{2n}$. *Low-dimensional similarities are computed using a heavy-tailed Student-t kernel:* $q_{ij} \propto (1 + \|y_i - y_j\|^2)^{-1}$. *The loss minimizes the KL divergence:*

$$\mathcal{L}_{tSNE} = \sum_{i \neq j} p_{ij} \log \frac{p_{ij}}{q_{ij}}.$$

**Definition A.2** *UMAP defines* $p_{j|i} = \exp(-(\|x_i - x_j\| - \rho_i)/\tau_i)$ *using adaptive exponential kernels, where* $\rho_i$ *is the local connectivity threshold. Symmetrized* $p_{ij}$ *is computed via fuzzy set union. In the embedding space,* $q_{ij} = (1 + a\|y_i - y_j\|^2)^{-b}$ *with fixed parameters* $(a, b)$. *The loss is a weighted binary cross-entropy:*

$$\mathcal{L}_{UMAP} = \sum_{i \neq j} \left[ p_{ij} \log \frac{p_{ij}}{q_{ij}} + (1 - p_{ij}) \log \frac{1 - p_{ij}}{1 - q_{ij}} \right].$$

## A.2 Contrastive Neighbor Embedding (CNE).

**Definition A.3** *Given a kNN graph, high-dimensional similarities are binary:* $S_{ij}^{d_h} = 1$ *if* $x_j \in kNN(x_i)$, *and* 0 *otherwise. In the embedding space, similarities are defined using a Cauchy kernel:* $S_{ij}^{d_l} = \phi(\mathbf{y}_i, \mathbf{y}_j) = \frac{1}{1+\|\mathbf{y}_i - \mathbf{y}_j\|^2}$. *The CNE objective combines attractive and repulsive forces:*

$$\mathcal{L}(\theta) = -\mathbb{E}_{(i,j)\sim p_i} \log \phi(f_\theta(\mathbf{x}_i), f_\theta(\mathbf{x}_j)) - b\mathbb{E}_{(i,j)} \log(1 - \phi(f_\theta(\mathbf{x}_i), f_\theta(\mathbf{x}_j))),$$

*where* $p_i$ *samples positive pairs and* $b > 0$ *balances the repulsion term.*

## A.3 Hyperbolic Models and Distance Calculation.

There are several equivalent models of hyperbolic geometry exist, including the Poincaré ball model, lorentz model (or hyperboloid model) and the upper half-space model. The mathematical framework of the $d$-dimensional hyperboloid model of hyperbolic geometry is deined as follows:

For $x, y \in \mathbb{R}^{d+1}$, the Lorentz product is an indefinite inner product given by,

$$x \circ y := x_1 y_1 - (x_2 y_2 + \cdots + x_{d+1} y_{d+1}). \tag{13}$$

The real vector space $\mathbb{R}^{d+1}$ equipped with this inner product is called *Lorentz space*, denoted by $\mathbb{R}^{1,d}$. It contains the *positive Lorentz space* as a subset:

$$\mathbb{R}_+^{1,d} := \left\{ x \in \mathbb{R}^{1,d} \ : \ x_1 > 0 \right\}.$$

Within $\mathbb{R}_+^{1,d}$, the *single-sheet hyperboloid* $\mathbb{H}^{d_h}$ is given by

$$\mathbb{H}^{d_h} := \left\{ x \in \mathbb{R}^{1,d} \ : \ x \circ x = 1, \ x_1 > 0 \right\}. \tag{14}$$

The hyperboloid model in dimension $d$ with curvature $-\kappa$ (for $\kappa > 0$) consists of $\mathbb{H}^{d_h}$ endowed with the hyperbolic distance:

$$d_{\mathbb{H}}^\kappa(x, y) = \frac{1}{\sqrt{\kappa}} \text{arcosh}(x \circ y), \quad x, y \in \mathbb{H}^{d_h}. \tag{15}$$

The distance $d_{\mathbb{H}}^\kappa$ is a valid metric on $\mathbb{H}^{d_h}$, it is positive definite and satisfies the triangle inequality. Moreover, equipped with the metric tensor:

$$ds^2 = \frac{1}{\kappa}(dx \circ dx),$$

the hyperboloid $\mathbb{H}^{d_h}$ becomes a Riemannian manifold of constant sectional curvature $-\kappa$, and $d_{\mathbb{H}}^\kappa$ corresponds exactly to its geodesic distance. In particular, the curvature $\kappa$ does not alter the definition of the manifold $\mathbb{H}^{d_h}$ itself, but only scales the distance metric. Just as Euclidean space is the canonical model for zero curvature, hyperbolic space is the canonical geometry for constant negative curvature.

### A.3.1 Poincaré Ball Model.

The Poincaré ball model is the most widely used formulation of hyperbolic space in machine learning [19, 40]. It defines the $n$-dimensional hyperbolic space as $\mathbb{B}^n = \{x \in \mathbb{R}^n : \|x\| < 1\}$ with Riemannian metric $g_x = \left(\frac{2}{1-\|x\|^2}\right)^2 I_n$. The hyperbolic distance between two points $u, v \in \mathbb{B}^n$ is:

$$d_{\mathbb{B}^n}(u, v) = \operatorname{arcosh}\left(1 + \frac{2\|u - v\|^2}{(1 - \|u\|^2)(1 - \|v\|^2)}\right). \tag{16}$$

This distance increases exponentially near the boundary, enabling natural hierarchical embeddings where central points correspond to root nodes and peripheral points to leaves.

### A.4 CO-SNE

**Definition A.4** *CO-SNE defines the similarities via hyperbolic normal kernels in the high-dimensional Poincaré ball* $\mathbb{B}^n$: $p_{j|i} = \exp\left(-d_{\mathbb{B}^n}(x_i, x_j)^2/2\sigma_i^2\right)/Z_i$, *with* $p_{ij} = (p_{j|i} + p_{i|j})/2m$. *In the embedding space* $\mathbb{B}^2$, *similarities use a hyperbolic Cauchy kernel:* $q_{ij} = \gamma^2/(d_{\mathbb{B}^2}(y_i, y_j)^2 + \gamma^2)/Z$. *The loss combines KL divergence with a norm-based regularizer:*

$$\mathcal{L}_{CO\text{-}SNE} = \lambda_1 \sum_{i,j} p_{ij} \log \frac{p_{ij}}{q_{ij}} + \lambda_2 \sum_i (\|x_i\|^2 - \|y_i\|^2)^2. \tag{17}$$

### A.5 Classical MDS

Utilizing the measurements of distances among pairs of objects, MDS (multidimensional scaling) finds a representation of each object in $d$ - dimensional space such that the distances are preserved in the estimated configuration as closely as possible. To validate the goodness-of-fit measure, MDS optimizes the loss function (known as "Stress"$(\sigma)$) given by:

$$\sigma(X) = \min_X \sum_{i < j \leq N} w_{ij} \left(\delta_{ij} - d_{ij}(X)\right)^2, \tag{18}$$

, where the observation mask is $W$ where $w_{ij} = 1$ if the distance $\delta_{ij}$ is known and $w_{ij} = 0$ otherwise, with the block structure:

$$W = \begin{bmatrix} \mathbf{0}_{N \times N} & \mathbf{1}_{N \times M} \\ \mathbf{1}_{M \times N}^\top & \mathbf{1}_{M \times M} \end{bmatrix} \tag{19}$$

where $\mathbf{0}$ and $\mathbf{1}$ denote matrices of zeros and ones, respectively and $X$ represents the computed configuration, $d_{ij}(X) = \|\boldsymbol{x_i} - \boldsymbol{x_j}\|$ is the Euclidean distance between nodes $i$ and $j$, $\delta_{ij}$ is the measured distance computed privately. Placing the weights of unknown inter-user distance to zero, the weight matrix $W$ can be partitioned into block matrices as shown in 19, where $11_{N,M}$ is a matrix of ones with shape $N \times M$. De Leeuw [13] applied an iterative method called SMACOF (Scaling by Majorizing a Convex Function) to estimate the configuration $X$. As the objective is a non-convex function, SMACOF minimizes the stress using the simple quadratic function $\tau(X, Z)$ which bounds $\sigma(X)$ (the complicated function) from above and meets the surface at the so-called supporting point $Z$ as defined below:

$$\sigma(X) \leq \tau(X, Z) = \sum_{i<j} w_{ij}\delta_{ij}^2 + \sum_{i<j} w_{ij}d_{ij}^2(X) - 2\sum_{i<j} w_{ij}\delta_{ij}^2 \frac{(\boldsymbol{x_i} - \boldsymbol{x_j})^T (\boldsymbol{z_i} - \boldsymbol{z_j})}{\|\boldsymbol{z_i} - \boldsymbol{z_j}\|} \tag{20}$$

Equation (20) can be written in matrix form as:

$$\tau(X, Z) = C + \operatorname{tr}\left(X^T V X\right) - 2\operatorname{tr}\left(X^T B(Z)Z\right). \tag{21}$$

The iterative solution which guarantees monotone convergence of stress [12] is given by equation (22), where $Z = X^{k-1}$:

$$X^{(k)} = \min_X \tau(X, Z) = V^\dagger B(X^{(k-1)})X^{(k-1)} \tag{22}$$

This algorithm offers flexibility to embed features in any dimension other than $d$, which enables the handling of high-dimensional data and also meets privacy constraints. As $V$ is not of full rank, hence the Moore-Penrose pseudoinverse $V^\dagger$ is used. The elements of the matrix $B(X)$ and $V$ are defined

in equation (23).

$$
b_{ij} = \begin{cases} -\dfrac{w_{ij}\delta_{ij}}{d_{ij}(\mathbf{X})}, & \text{if } d_{ij}(\mathbf{X}) \neq 0, \ i \neq j \\ 0, & \text{if } d_{ij}(\mathbf{X}) = 0, \ i \neq j \\ -\sum\limits_{j=1,\ j\neq i}^{N} b_{ij}, & \text{if } i = j \end{cases}
\tag{23}
$$

$$
v_{ij} = \begin{cases} -w_{ij}, & \text{if } i \neq j \\ -\sum\limits_{j=1,\ j\neq i}^{N} v_{ij}, & \text{if } i = j \end{cases}
$$

## A.6 SENSE: Pseudocode

---
**Algorithm 1** SENSE Framework

---
**Require:** Anchors $\mathbf{X}_A \in \mathbb{R}^{K \times d_{\mathrm{h}}}$, client datasets $\{\mathcal{D}_m = \{x_i^m\}_{i=1}^{N_m}\}_{m=1}^{M}$, target dim $d_\ell$, high/low
    geometry $\mathbb{G}_{\mathrm{high}} \in \{\mathbb{R}^{d_{\mathrm{h}}}, \mathbb{H}^{d_{\mathrm{h}}}\}$, $\mathbb{G}_{\mathrm{low}} \in \{\mathbb{R}^{d_\ell}, \mathbb{H}^{d_\ell}\}$

**Ensure:** Global embeddings $\{\mathbf{Y}^m \in \mathbb{G}_{\mathrm{low}}^{N_m}\}_{m=1}^{M}$
  1: Server broadcasts $\mathbf{X}_A$ to all clients
  2: **for** each client $\mathcal{C}_m$ **do**
  3:     Compute distances $\mathbf{d}_i^m = \mathcal{D}_{\mathbb{G}_{\mathrm{high}}}(x_i^m, \mathbf{X}_A)$ for all $x_i^m \in \mathcal{D}_m$
  4:     Send $\{\mathbf{d}_i^m\}_{i=1}^{N_m}$ to server
  5: **end for**
  6: Server builds observed matrix $\mathbf{D}_\Omega$ using $E$, $F$, (optionally $G$)
  7: Complete $\widehat{\mathbf{D}}$ via structured matrix completion; extract $\widehat{\mathbf{G}}$
  8: Compute similarities $S^{d_{\mathrm{h}}}$ from $\widehat{\mathbf{G}}$ using kernel $f$ (see Eqns 6, 7)
  9: Learn embedding $\mathbf{Y}$ in $\mathbb{G}_{\mathrm{low}}$ using NE, contrastive, or CO-SNE objective

---

## A.7 SENSE via Anchored-MDS: Pseudocode

---
**Algorithm 2** SENSE via Anchored-MDS

---
**Require:** Anchor embeddings $X_A \in \mathbb{R}^{K \times d_h}$, observed entries $\mathcal{P}_\Omega(D)$, target dim $d_h$, tolerance $\epsilon$,
    max iterations $T$

**Ensure:** Reconstructed embeddings $X_{NA} \in \mathbb{R}^{N \times d_h}$
  1: Initialize $X_{NA}^{(0)}$ randomly, set $k \leftarrow 1$
  2: **while** $k \leq T$ **do**
  3:     Form $X^{(k-1)} = \begin{bmatrix} X_A & X_{NA}^{(k-1)} \end{bmatrix}^T$
  4:     Compute $\mathcal{P}_\Omega(D(X^{(k-1)}))$
  5:     Construct $W$ and compute $V$, $B(X^{(k-1)})$ respecting $\Omega$
  6:     Update $X_{NA}^{(k)}$ using Eq. (11)
  7:     If stress improvement $< \epsilon$, **break**; else $k \leftarrow k + 1$
  8: **end while**
  9: **return** $X_{NA}^{(k)}$

---

## A.8 Anchor Generation

In the proposed method, distribution of the anchor data is critical. The anchor is a common information shared between all the clients. The anchor data is generated randomly or by open data for securing privacy. The proper scheduling of the anchors has a significant impact on the overall performance and accuracy of the framework. There are several factors to consider when developing the anchor scheduling strategy, including:

**Number of anchors**: The number of anchors used in the framework has a direct impact on the algorithmic performance. Too few anchors may not preserve the structural information while ensuring privacy, while too many anchors may lead to overfitting and may violate privacy.

 **Selection criteria**: The criteria used to select anchors can also impact the performance of the system.
Selecting anchors from the same probability distribution as of the underlying user data may be more
effective than selecting them at random. For example, the data distribution of patient similarity
networks or social networks will depend on factors including a number of patients/users or similarity
of patients/connection between users.

Table 4: Observed index sets $\Omega$ used for SENSE under each client configuration. Here, $\mathcal{A}_G$ denotes global anchors, $\mathcal{A}_L^{(j)}$ are local anchors accessible only to client $j$, and $\mathcal{X}^{(m)}$ are NA indices at client $m$. Binary masks $W_F$ and $W_G$ indicate anchor-to-NA and intra-client NA–NA visibility. Observed distances are used to construct $V$, $B(X)$, and select relevant rows of $X_A$ for embedding computation.

| SENSE Setting | Observed Index Set $\Omega$ |
|---|---|
| **Pointwise-Full** | Each client holds one NA. All anchor-to-NA distances are known; no NA–NA or local anchor information. $\Omega = \{(i,j) : i \in \mathcal{A}_G, \ j \in [K+1, K+N]\} \cup \{(j,i) : i \in \mathcal{A}_G, \ j \in [K+1, K+N]\}$ |
| **Pointwise-Partial** | Each client holds one NA. Global anchors $\mathcal{A}_G$ are shared across all clients. Local anchors $\mathcal{A}_L^{(j)}$ are only accessible to client $j$. $\Omega = \bigcup_{j=1}^{N} \left( (\mathcal{A}_G \cup \mathcal{A}_L^{(j)}) \times \{K+j\} \cup \{K+j\} \times (\mathcal{A}_G \cup \mathcal{A}_L^{(j)}) \right)$ |
| **Multisite-Full** | Each client holds multiple NAs. All anchor-to-NA distances are known. Intra-client NA–NA distances are observed. $\Omega = \{(i,j) : i \in \mathcal{A}_G, j \in [K+1, K+N]\} \cup \{(j,i) : i \in \mathcal{A}_G, j \in [K+1, K+N]\} \cup \bigcup_{m=1}^{M}(\mathcal{X}^{(m)} \times \mathcal{X}^{(m)})$ |
| **Multisite-Partial** | Each client holds multiple NAs. Anchor-to-NA distances are partially known via $W_F$ (global + local anchors). Intra-client NA–NA distances are observed via $W_G$. $\Omega = \{(i, j+K) : W_F[i,j] = 1\} \cup \{(j+K, i) : W_F[i,j] = 1\} \cup \{(i,j) : W_G[i,j] = 1\}$ |

## A.9 Theoretical Proofs.

Unlike some EDG [52] methods that assume uniform random sampling of pairwise distances, SENSE
uses a structured sampling scheme where anchor-to-NA distances are measured by design. This
enables deterministic recovery guarantees based on geometric conditions (e.g., connectivity to affinely
independent anchors), avoiding reliance on probabilistic bounds from random sampling.

**Proof A.1** *Each NA point $\boldsymbol{x}_j \in \mathbb{R}^{d_h}$ computes squared distances to a subset of anchors indexed by $\mathcal{I}_j$, with $r_j = |\mathcal{I}_j|$. This yields $r_j$ quadratic constraints of the form:*

$$\|\boldsymbol{x}_j - \boldsymbol{a}_i\|^2 = d_{hij}^2, \quad \forall i \in \mathcal{I}_j.$$

*To analyze identifiability, fix a reference anchor $\boldsymbol{a}_k \in \mathcal{I}_G$ from the global anchor set, and consider the difference of equations relative to this reference:*

$$\|\boldsymbol{x}_j - \boldsymbol{a}_i\|^2 - \|\boldsymbol{x}_j - \boldsymbol{a}_k\|^2 = d_{hij}^2 - d_{hkj}^2.$$

*Expanding and simplifying yields the linear system:*

$$2(\boldsymbol{a}_k - \boldsymbol{a}_i)^\top \boldsymbol{x}_j = \|\boldsymbol{a}_k\|^2 - \|\boldsymbol{a}_i\|^2 + d_{hij}^2 - d_{hkj}^2, \quad \forall i \in \mathcal{I}_j \setminus \{k\}.$$

*Letting $A_j \in \mathbb{R}^{(r_j-1) \times d}$ denote the coefficient matrix and $\boldsymbol{b}_j$ the RHS vector, we write:*

$$A_j \boldsymbol{x}_j = \boldsymbol{b}_j.$$

*This is a system of $r_j - 1$ linear equations in $d_h$ unknowns. If $r_j < d_h + 1$, then $rank(A_j) \leq r_j - 1 < d_h$, and the solution set $\{\boldsymbol{x}_j \in \mathbb{R}^{d_h} : A_j \boldsymbol{x}_j = \boldsymbol{b}_j\}$ forms an affine subspace of dimension at least $d_h - r_j + 1$. Hence, infinitely many solutions exist that satisfy the same anchor distances, preventing exact recovery of $\boldsymbol{x}_j$.*

*To ensure privacy across all clients (both pointwise and multisite), we enforce:*

$$|\mathcal{I}_j| = K_G + K_L^{(j)} \leq d_h, \quad \forall j \in [N],$$

606 where $K_L^{(j)}$ is the number of local anchors accessible to $\boldsymbol{x}_j$. In the multisite case, local anchors
607 are restricted to the corresponding client, and global anchors are common across all clients. This
608 structure ensures that even with partial anchor visibility, each client's feature vector cannot be
609 uniquely recovered from its observed distances.

610 **Remark 3** Each anchor distance imposes a quadratic constraint on the unknown $\boldsymbol{x}_j \in \mathbb{R}^{d_h}$. If the
611 number of constraints $r_j$ is less than the ambient dimension $d$, the system is underdetermined and
612 has infinitely many solutions. Thus, SENSE preserves privacy by bounding the number of anchor
613 distances accessible to each client.

614 **Proof A.2** Consider a network in $d_h$-dimensional Euclidean space $\mathbb{R}^{d_h}$, comprising anchors $A =$
615 $\{A_1, A_2, \ldots, A_K\}$ and non-anchor nodes $P = \{P_1, P_2, \ldots, P_N\}$, with feature vectors $\boldsymbol{x_i} \in \mathbb{R}^{d_h}$.
616 Anchors locations are known, while non-anchors need estimation. Previous work [31] shows that in
617 $\mathbb{R}^{d_h}$, a minimum of $(d+1)$ anchors with known locations is required to locate $N$ non-anchor nodes.
618 The utilization of anchors for distributed sensor localization constitutes a thoroughly investigated
619 domain, underpinned by the following assumptions:

620 • **(A1)** Non-anchor nodes lie inside the convex hull of the anchors, i.e., $C(P) \subseteq C(A)$.
621 • **(A2)** Each non-anchor node $P_i$ has at least one set of neighbor nodes $N_i \subset (A \cup P)$ with
622   $|N_i| = d_h + 1$ such that $i$ lies inside $C(N_i)$.
623 • **(A3)** In the set $\{i \cup N_i\}$, every non-anchor node $i$ can obtain the inter-node distances among all
624   nodes.

625 However, to accurately recover features in $\mathbb{R}^{d_h}$, at least $d_h$ anchors are necessary, even if non-anchors
626 are placed in any location. Thus, having fewer than $d_h$ anchors, i.e., $K < d_h$, guarantees that exact
627 feature embeddings cannot be obtained, ensuring privacy.

628 **Proof A.3** From Theorem 3.1 (Exact Recovery) in [30], the L-HYDRA algorithm guarantees recovery
629 up to isometry only if $K \geq d_h$ and the $K$ anchors are in general position (not lying on a single
630 hyperbolic hyperplane). If $K < d_h$, then the system of equations defined by $E$ and $F$ is underdeter-
631 mined: the landmarks do not span $\mathbb{H}_h^{d_h}$, and multiple embeddings of the NA points are consistent
632 with the observed distances. Hence, SENSE ensures privacy by choosing $K < d_h$, preventing unique
633 reconstruction of private client embeddings.

634 **A.10 Metric Used.**
635 • *Cosine Similarity (*CosSim*):* Measures angular similarity between the original NA feature matrix
636   $X'_{\text{NA}} \in \mathbb{R}^{N \times d_h}$ and the reconstructed version $X_{\text{NA}} \in \mathbb{R}^{N \times d_h}$ from SENSE-anchored MDS. Cosine
637   similarity is computed as:

$$\text{CosSim}(X'_{\text{NA}}, X_{\text{NA}}) = \frac{1}{N} \sum_{i=1}^{N} \frac{\langle X'_{\text{NA}}{}^{(i)}, X_{\text{NA}}^{(i)} \rangle}{\|X'_{\text{NA}}{}^{(i)}\| \cdot \|X_{\text{NA}}^{(i)}\|}$$

638 High values (close to 1) indicate strong alignment between original and reconstructed embeddings.
639 • *Distance Error (DE):* and *F-score (FS):* defined in Section 4.1.
640 • *Pearson Correlation ($\rho$):* Quantifies linear correlation between the original and reconstructed
641   NA–NA distance matrices:

$$\rho = \text{Pearson}(G_{ij}, \widehat{G}_{ij}), \quad \forall i < j$$

642 where $G$ and $\widehat{G}$ denote the ground-truth and reconstructed distance matrices respectively. Values
643 close to 1 indicate that the relative distance structure is preserved.
644 • *Frobenius Norm Error ($X_{frob}$):* Measures reconstruction error in the embedding space:

$$X_{\text{frob}} = \frac{\|X_{\text{NA}} - X'_{\text{NA}}\|_F}{\|X'_{\text{NA}}\|_F}$$

645 A value of 0 implies perfect reconstruction; higher values suggest increasing deviation.

## A.11 Dataset Statistics.

Table 5: Dataset statistics and learning setups grouped by embedding geometry. For hyperbolic, the stats are for *Pointwise* setting.

| Space | Dataset | #Classes | #Datapoints | #Clients (M) | Dimension |
|---|---|---|---|---|---|
| | MNIST | 10 | 25000 | 10 | 784 |
| | Fashion-MNIST | 10 | 25000 | 10 | 784 |
| | CIFAR-10 | 10 | 25000 | 5/10 | 1024 |
| | DermaMNIST | 7 | 10015 | 10 | 784 |
| | PneumoniaMNIST | 2 | 5856 | 10 | 784 |
| Euclidean | RetinaMNIST | 5 | 1600 | 10 | 784 |
| | BreastMNIST | 2 | 780 | 10 | 784 |
| | BloodMNIST | 8 | 17092 | 10 | 784 |
| | OrganCMNIST | 11 | 23583 | 10 | 784 |
| | OrganSMNIST | 11 | 25211 | 10 | 784 |
| | German-Credit | 2 | 1000 | 10 | 20 |
| | Airport | 4 | 3185 | 3185 | 11 |
| Hyperbolic | Amazon | - | 5000 | 5000 | 128 |
| | DBLP | - | 5000 | 5000 | 128 |

## A.12 System Specifications

All experiments are conducted on a server equipped with two **NVIDIA RTX A6000** GPUs (48 GB memory each) and an **Intel Xeon Platinum 8360Y** CPU with **1 TB RAM**.

## A.13 Visualization Results

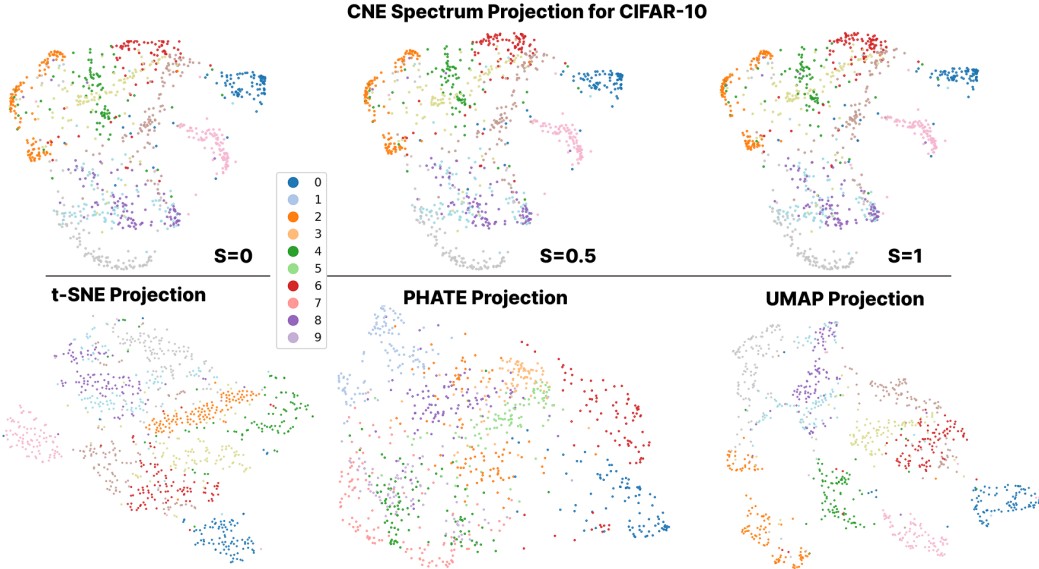

Figure 4: Pointwise setting: CIFAR-10 (1000 non-anchor points, 783 anchors)

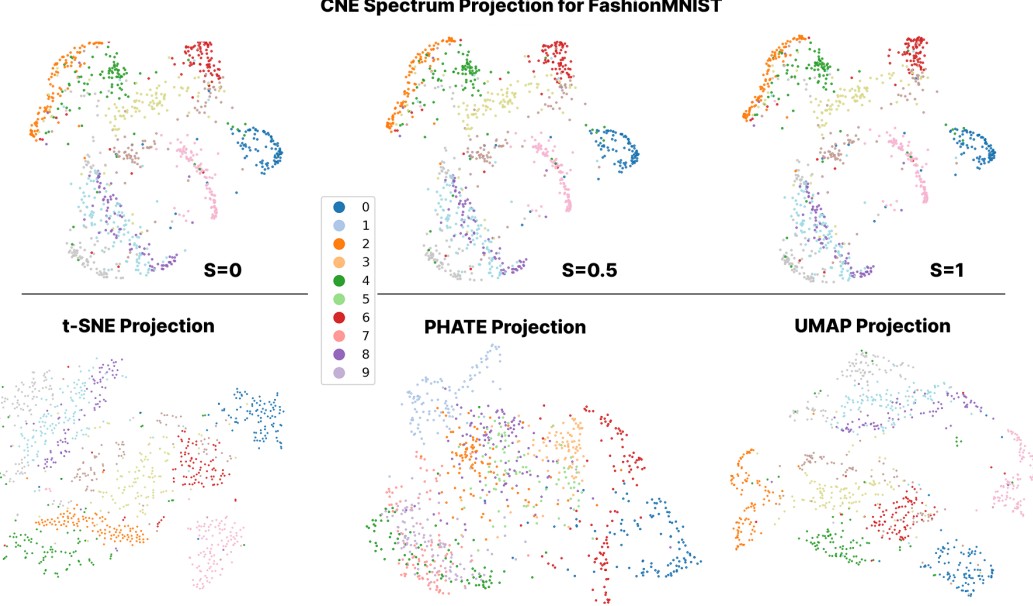

Figure 5: Pointwise setting: FashionMNIST (1000 non-anchor points, 783 anchors)

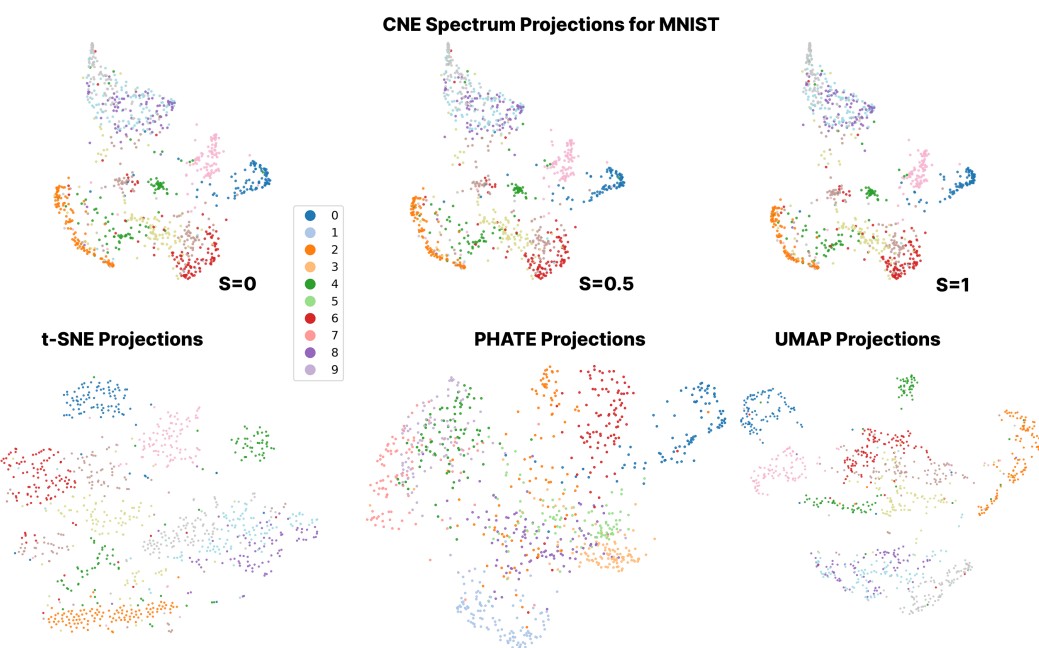

Figure 6: Pointwise setting: MNIST (1000 non-anchor points, 783 anchors)

 **A.14 Results.**

Table 6: FS and DE across IID, and non-IID balanced and unbalanced splits.

| Data | IID | | Bal | | Unbal | |
|---|---|---|---|---|---|---|
| | FS | DE | FS | DE | FS | DE |
| PNEU. | 0.92 | 0.0052 | 0.87 | 0.0066 | 0.91 | 0.0055 |
| BLOOD | 0.90 | 0.0052 | 0.89 | 0.0051 | 0.90 | 0.0052 |
| BREAST | 0.95 | 0.0092 | 0.92 | 0.0113 | 0.91 | 0.0124 |
| DERMA | 0.96 | 0.0029 | 0.93 | 0.0031 | 0.96 | 0.0029 |
| RETINA | 0.96 | 0.0221 | 0.94 | 0.0272 | 0.96 | 0.0214 |
| ORGANC | 0.80 | 0.0092 | 0.79 | 0.0089 | 0.79 | 0.0092 |
| ORGANS | 0.81 | 0.0089 | 0.80 | 0.0085 | 0.81 | 0.0093 |
| GERMAN | 0.75 | 0.0565 | 0.73 | 0.0621 | 0.72 | 0.0629 |

Table 7: FS and DE under POINTWISE, IID, and NON-IID settings, comparing MULTISITE-FULL and MULTISITE-PARTIAL.

| Dataset | Pointwise | | IID-Full | | IID-Partial | | Non-IID-Full | | Non-IID-Partial | |
|---|---|---|---|---|---|---|---|---|---|---|
| | FS | DE | FS | DE | FS | DE | FS | DE | FS | DE |
| MNIST | 0.9557 | 0.0057 | 0.8034 | 0.0097 | 0.9266 | 0.0438 | 0.7864 | 0.0101 | 0.9275 | 0.0434 |
| FashionMNIST | 0.9560 | 0.0058 | 0.7586 | 0.0070 | 0.8726 | 0.0153 | 0.7534 | 0.0070 | 0.8754 | 0.0156 |
| CIFAR-10 | 0.9562 | 0.0057 | 0.9303 | 0.0049 | 0.9277 | 0.0044 | 0.9308 | 0.0049 | 0.9380 | 0.0044 |

Table 8: Multisite setting comparison Non-iid unbalanced: Full vs Partial: Evaluation of different methods (Vanilla and SENSE variants) across different metrics.

| Data | Metric | t-SNE | | UMAP | | PHATE | | CNE(s=0) | | CNE(s=0.5) | | CNE(s=1) | |
|---|---|---|---|---|---|---|---|---|---|---|---|---|---|
| | | VAN. | SENSE | VAN. | SENSE | VAN. | SENSE | VAN. | SENSE | VAN. | SENSE | VAN. | SENSE |
| | | | | | | — Multisite-Partial Setting — | | | | | | | |
| | Trust. | 0.9259 | 0.9274 | 0.7447 | 0.7476 | 0.8175 | 0.8174 | 0.8334 | 0.8336 | 0.8322 | 0.8321 | 0.8232 | 0.8244 |
| CIFAR-10 | Cont. | 0.9107 | 0.9391 | 0.8756 | 0.8804 | 0.9369 | 0.9381 | 0.9554 | 0.9552 | 0.9552 | 0.9549 | 0.9565 | 0.9561 |
| | Stead. | 0.8099 | 0.8165 | 0.6904 | 0.6938 | 0.7363 | 0.7349 | 0.7609 | 0.7654 | 0.7619 | 0.7580 | 0.7415 | 0.7487 |
| | Cohes. | 0.4707 | 0.4806 | 0.3725 | 0.3752 | 0.4927 | 0.4857 | 0.4708 | 0.4630 | 0.4716 | 0.4778 | 0.4766 | 0.4793 |
| | | | | | | — Multisite-Full Setting — | | | | | | | |
| | Trust. | 0.9259 | 0.9270 | 0.7447 | 0.7482 | 0.8175 | 0.8168 | 0.8334 | 0.8336 | 0.8322 | 0.8329 | 0.8232 | 0.8247 |
| CIFAR-10 | Cont. | 0.9107 | 0.9364 | 0.8756 | 0.8808 | 0.9369 | 0.9366 | 0.9554 | 0.9553 | 0.9552 | 0.9550 | 0.9565 | 0.9561 |
| | Stead. | 0.8099 | 0.8229 | 0.6904 | 0.6875 | 0.7363 | 0.7357 | 0.7609 | 0.7624 | 0.7619 | 0.7580 | 0.7415 | 0.7464 |
| | Cohes. | 0.4707 | 0.4673 | 0.3725 | 0.3674 | 0.4927 | 0.4831 | 0.4708 | 0.4662 | 0.4716 | 0.4690 | 0.4766 | 0.4811 |
| | | | | | | — Pointwise-Full Setting — | | | | | | | |
| | Trust. | 0.9683 | 0.9659 | 0.9435 | 0.9419 | 0.8488 | 0.8531 | 0.9112 | 0.9123 | 0.9082 | 0.9079 | 0.9021 | 0.9035 |
| | Cont. | 0.9465 | 0.9448 | 0.9379 | 0.9333 | 0.9533 | 0.9527 | 0.9446 | 0.9442 | 0.9458 | 0.9437 | 0.9445 | 0.9442 |
| CIFAR-10 | Stead. | 0.8061 | 0.8081 | 0.7793 | 0.7825 | 0.7111 | 0.7165 | 0.7992 | 0.7878 | 0.7887 | 0.8005 | 0.7808 | 0.7920 |
| | Cohes. | 0.7482 | 0.7672 | 0.7415 | 0.7336 | 0.7431 | 0.7365 | 0.7485 | 0.7451 | 0.7513 | 0.7473 | 0.7435 | 0.7350 |

Table 9: IID setting: Evaluation of different dimensionality reduction methods (Vanilla and SENSE variants) across various metrics.

| Data | Metric | t-SNE | | UMAP | | PHATE | | CNE(s=0) | | CNE(s=0.5) | | CNE(s=1) | |
|---|---|---|---|---|---|---|---|---|---|---|---|---|---|
| | | VAN. | SENSE | VAN. | SENSE | VAN. | SENSE | VAN. | SENSE | VAN. | SENSE | VAN. | SENSE |
| PneumoniaMNIST | Trust. | 0.9718 | 0.9700 | 0.7687 | 0.7700 | 0.8573 | 0.8590 | 0.9016 | 0.9026 | 0.8973 | 0.8967 | 0.8837 | 0.8795 |
| | Cont. | 0.9395 | 0.9442 | 0.9145 | 0.9143 | 0.9616 | 0.9598 | 0.9592 | 0.9587 | 0.9591 | 0.9582 | 0.9606 | 0.9598 |
| | Stead. | 0.7840 | 0.7844 | 0.6203 | 0.6272 | 0.7158 | 0.7228 | 0.7554 | 0.7516 | 0.7439 | 0.7424 | 0.7369 | 0.7263 |
| | Cohes. | 0.7031 | 0.6963 | 0.6081 | 0.6272 | 0.6902 | 0.6898 | 0.7013 | 0.7112 | 0.6981 | 0.6970 | 0.7006 | 0.7050 |
| BloodMNIST | Trust. | 0.9628 | 0.9611 | 0.8643 | 0.8633 | 0.8515 | 0.8527 | 0.8847 | 0.8820 | 0.8793 | 0.8820 | 0.8729 | 0.8736 |
| | Cont. | 0.9312 | 0.9280 | 0.9416 | 0.9391 | 0.9444 | 0.9440 | 0.9555 | 0.9558 | 0.9556 | 0.9558 | 0.9553 | 0.9556 |
| | Stead. | 0.7515 | 0.7436 | 0.6899 | 0.6764 | 0.6967 | 0.6871 | 0.7259 | 0.7211 | 0.7228 | 0.7211 | 0.7164 | 0.7133 |
| | Cohes. | 0.7085 | 0.7106 | 0.7233 | 0.7261 | 0.7416 | 0.7469 | 0.7435 | 0.7339 | 0.7329 | 0.7339 | 0.7453 | 0.7462 |
| BreastMNIST | Trust. | 0.9382 | 0.9370 | 0.7599 | 0.7589 | 0.8835 | 0.8774 | 0.8938 | 0.8924 | 0.8939 | 0.8920 | 0.8934 | 0.8924 |
| | Cont. | 0.9452 | 0.9412 | 0.8147 | 0.8174 | 0.9533 | 0.9526 | 0.9450 | 0.9446 | 0.9450 | 0.9445 | 0.9450 | 0.9444 |
| | Stead. | 0.8522 | 0.8514 | 0.5800 | 0.5697 | 0.8056 | 0.8099 | 0.8400 | 0.8400 | 0.8287 | 0.8308 | 0.8317 | 0.8353 |
| | Cohes. | 0.6028 | 0.5987 | 0.4226 | 0.4226 | 0.5639 | 0.5611 | 0.5566 | 0.5605 | 0.5637 | 0.5670 | 0.5532 | 0.5606 |
| DermaMNIST | Trust. | 0.9758 | 0.9762 | 0.7513 | 0.7480 | 0.8726 | 0.8726 | 0.9129 | 0.9118 | 0.9125 | 0.9126 | 0.9017 | 0.9023 |
| | Cont. | 0.9592 | 0.9583 | 0.9134 | 0.9129 | 0.9736 | 0.9729 | 0.9709 | 0.9712 | 0.9707 | 0.9706 | 0.9716 | 0.9714 |
| | Stead. | 0.7995 | 0.7976 | 0.5930 | 0.5945 | 0.7332 | 0.7291 | 0.7726 | 0.7739 | 0.7694 | 0.7638 | 0.7580 | 0.7577 |
| | Cohes. | 0.7294 | 0.7107 | 0.5590 | 0.5618 | 0.7001 | 0.7184 | 0.7339 | 0.7334 | 0.7390 | 0.7373 | 0.7308 | 0.7297 |
| RetinaMNIST | Trust. | 0.9797 | 0.9758 | 0.8777 | 0.8643 | 0.9144 | 0.9038 | 0.9480 | 0.9335 | 0.9469 | 0.9331 | 0.9450 | 0.9313 |
| | Cont. | 0.9669 | 0.9567 | 0.9280 | 0.9232 | 0.9738 | 0.9730 | 0.9718 | 0.9711 | 0.9704 | 0.9700 | 0.9678 | 0.9678 |
| | Stead. | 0.8483 | 0.8479 | 0.6120 | 0.5941 | 0.7618 | 0.7434 | 0.8183 | 0.8140 | 0.8117 | 0.8050 | 0.8105 | 0.8086 |
| | Cohes. | 0.7051 | 0.6963 | 0.5835 | 0.5515 | 0.6980 | 0.6995 | 0.7123 | 0.7074 | 0.7046 | 0.7112 | 0.6831 | 0.7135 |
| OrganCMNIST | Trust. | 0.9608 | 0.9482 | 0.8879 | 0.8815 | 0.8845 | 0.8858 | 0.9149 | 0.9028 | 0.9160 | 0.9039 | 0.9024 | 0.8890 |
| | Cont. | 0.9238 | 0.9413 | 0.9231 | 0.9242 | 0.9696 | 0.9682 | 0.9731 | 0.9683 | 0.9730 | 0.9679 | 0.9738 | 0.9688 |
| | Stead. | 0.6948 | 0.8027 | 0.7575 | 0.7678 | 0.7994 | 0.8058 | 0.8690 | 0.8677 | 0.8788 | 0.8673 | 0.8624 | 0.8593 |
| | Cohes. | 0.4762 | 0.4849 | 0.3335 | 0.3145 | 0.5695 | 0.5153 | 0.4751 | 0.4760 | 0.5268 | 0.5001 | 0.5545 | 0.5166 |
| OrganSMNIST | Trust. | 0.9565 | 0.9421 | 0.8707 | 0.8588 | 0.8766 | 0.8890 | 0.9130 | 0.9026 | 0.9128 | 0.9034 | 0.8991 | 0.8911 |
| | Cont. | 0.9219 | 0.9366 | 0.9248 | 0.9211 | 0.9679 | 0.9717 | 0.9741 | 0.9684 | 0.9732 | 0.9672 | 0.9737 | 0.9679 |
| | Stead. | 0.6793 | 0.7753 | 0.7305 | 0.7513 | 0.7786 | 0.7965 | 0.8609 | 0.8691 | 0.8649 | 0.8745 | 0.8517 | 0.8601 |
| | Cohes. | 0.4856 | 0.4702 | 0.3327 | 0.3316 | 0.5575 | 0.5094 | 0.4838 | 0.4525 | 0.5312 | 0.4889 | 0.5564 | 0.4783 |
| german-credit | Trust. | 0.9771 | 0.9553 | 0.9505 | 0.9330 | 0.8559 | 0.8551 | 0.9380 | 0.9224 | 0.9359 | 0.9140 | 0.9325 | 0.9192 |
| | Cont. | 0.9590 | 0.9434 | 0.9587 | 0.9449 | 0.9482 | 0.9294 | 0.9573 | 0.9448 | 0.9573 | 0.9429 | 0.9564 | 0.9432 |
| | Stead. | 0.8603 | 0.8251 | 0.8342 | 0.7907 | 0.7500 | 0.7228 | 0.8414 | 0.7954 | 0.8416 | 0.7883 | 0.8401 | 0.7944 |
| | Cohes. | 0.6810 | 0.6895 | 0.6542 | 0.6413 | 0.6712 | 0.6640 | 0.6465 | 0.6651 | 0.6577 | 0.6675 | 0.6624 | 0.6550 |

Table 10: Non-IID (balanced) setting: Evaluation of different methods (Vanilla and SENSE variants) across different metrics.

| Data | Metric | t-SNE | | UMAP | | PHATE | | CNE(s=0) | | CNE(s=0.5) | | CNE(s=1) | |
|---|---|---|---|---|---|---|---|---|---|---|---|---|---|
| | | VAN. | SENSE | VAN. | SENSE | VAN. | SENSE | VAN. | SENSE | VAN. | SENSE | VAN. | SENSE |
| PneumoniaMNIST | Trust. | 0.9566 | 0.9483 | 0.8806 | 0.8658 | 0.8909 | 0.8937 | 0.9430 | 0.9393 | 0.9372 | 0.9343 | 0.9226 | 0.9168 |
| | Cont. | 0.9228 | 0.9278 | 0.9031 | 0.9114 | 0.9776 | 0.9732 | 0.9683 | 0.9678 | 0.9690 | 0.9686 | 0.9704 | 0.9695 |
| | Stead. | 0.6952 | 0.7165 | 0.6007 | 0.6211 | 0.7146 | 0.7244 | 0.7778 | 0.7737 | 0.7694 | 0.7692 | 0.7622 | 0.7579 |
| | Cohes. | 0.6377 | 0.6815 | 0.6205 | 0.6070 | 0.6650 | 0.6771 | 0.7259 | 0.7162 | 0.7240 | 0.7145 | 0.7172 | 0.7336 |
| BloodMNIST | Trust. | 0.9304 | 0.9292 | 0.8902 | 0.8796 | 0.8640 | 0.8633 | 0.9003 | 0.8972 | 0.8959 | 0.8944 | 0.8862 | 0.8856 |
| | Cont. | 0.9020 | 0.9029 | 0.9385 | 0.9390 | 0.9510 | 0.9492 | 0.9618 | 0.9611 | 0.9620 | 0.9614 | 0.9622 | 0.9614 |
| | Stead. | 0.7060 | 0.7017 | 0.6815 | 0.6927 | 0.6812 | 0.6927 | 0.7531 | 0.7505 | 0.7466 | 0.7442 | 0.7536 | 0.7395 |
| | Cohes. | 0.6781 | 0.6761 | 0.7210 | 0.7096 | 0.7620 | 0.7540 | 0.7441 | 0.7603 | 0.7472 | 0.7335 | 0.7561 | 0.7603 |
| BreastMNIST | Trust. | 0.9643 | 0.9657 | 0.8476 | 0.8562 | 0.9188 | 0.9241 | 0.9403 | 0.9422 | 0.9385 | 0.9418 | 0.9383 | 0.9415 |
| | Cont. | 0.9632 | 0.9658 | 0.8567 | 0.8408 | 0.9587 | 0.9671 | 0.9604 | 0.9594 | 0.9598 | 0.9590 | 0.9599 | 0.9591 |
| | Stead. | 0.8331 | 0.8370 | 0.5159 | 0.5081 | 0.7585 | 0.7913 | 0.8712 | 0.8742 | 0.8684 | 0.8616 | 0.8691 | 0.8675 |
| | Cohes. | 0.6174 | 0.6018 | 0.3677 | 0.3741 | 0.5187 | 0.5165 | 0.5254 | 0.5667 | 0.5265 | 0.5413 | 0.5200 | 0.5485 |
| DermaMNIST | Trust. | 0.9545 | 0.9467 | 0.8253 | 0.8048 | 0.8963 | 0.8961 | 0.9335 | 0.9351 | 0.9292 | 0.9327 | 0.9147 | 0.9167 |
| | Cont. | 0.9403 | 0.9284 | 0.8977 | 0.8895 | 0.9825 | 0.9815 | 0.9742 | 0.9734 | 0.9743 | 0.9733 | 0.9761 | 0.9756 |
| | Stead. | 0.7304 | 0.7148 | 0.5608 | 0.5428 | 0.7327 | 0.7295 | 0.7901 | 0.7909 | 0.7834 | 0.7841 | 0.7751 | 0.7743 |
| | Cohes. | 0.6493 | 0.6484 | 0.5159 | 0.5152 | 0.6867 | 0.6726 | 0.6993 | 0.6976 | 0.6976 | 0.7128 | 0.6902 | 0.7012 |
| RetinaMNIST | Trust. | 0.9749 | 0.9743 | 0.8933 | 0.8829 | 0.9228 | 0.9227 | 0.9522 | 0.9523 | 0.9492 | 0.9519 | 0.9497 | 0.9495 |
| | Cont. | 0.9627 | 0.9616 | 0.9289 | 0.9152 | 0.9752 | 0.9729 | 0.9720 | 0.9713 | 0.9712 | 0.9700 | 0.9670 | 0.9675 |
| | Stead. | 0.8447 | 0.8380 | 0.6155 | 0.6174 | 0.7534 | 0.7559 | 0.8224 | 0.8172 | 0.8134 | 0.8189 | 0.8123 | 0.8046 |
| | Cohes. | 0.7140 | 0.7283 | 0.5785 | 0.5648 | 0.7189 | 0.6836 | 0.7292 | 0.7005 | 0.7092 | 0.6938 | 0.7039 | 0.6849 |
| OrganCMNIST | Trust. | 0.9489 | 0.9271 | 0.8975 | 0.8888 | 0.9005 | 0.8984 | 0.9235 | 0.9132 | 0.9232 | 0.9126 | 0.9140 | 0.8994 |
| | Cont. | 0.9210 | 0.9082 | 0.9232 | 0.9185 | 0.9737 | 0.9719 | 0.9756 | 0.9715 | 0.9750 | 0.9710 | 0.9760 | 0.9717 |
| | Stead. | 0.6365 | 0.7142 | 0.7462 | 0.7290 | 0.8038 | 0.7909 | 0.8611 | 0.8724 | 0.8660 | 0.8745 | 0.8621 | 0.8640 |
| | Cohes. | 0.4862 | 0.4913 | 0.3249 | 0.3191 | 0.5088 | 0.5154 | 0.5338 | 0.4980 | 0.5266 | 0.4974 | 0.4908 | 0.5282 |
| OrganSMNIST | Trust. | 0.9383 | 0.9093 | 0.8954 | 0.8861 | 0.9054 | 0.9071 | 0.9269 | 0.9190 | 0.9291 | 0.9194 | 0.9172 | 0.9092 |
| | Cont. | 0.9164 | 0.8881 | 0.9168 | 0.9255 | 0.9774 | 0.9758 | 0.9796 | 0.9746 | 0.9786 | 0.9741 | 0.9788 | 0.9741 |
| | Stead. | 0.5896 | 0.6154 | 0.6315 | 0.6953 | 0.7784 | 0.7963 | 0.8591 | 0.8684 | 0.8560 | 0.8634 | 0.8411 | 0.8523 |
| | Cohes. | 0.5109 | 0.5108 | 0.3441 | 0.3665 | 0.5642 | 0.5278 | 0.5079 | 0.4878 | 0.5461 | 0.5021 | 0.5487 | 0.5001 |
| german-credit | Trust. | 0.9752 | 0.9575 | 0.9511 | 0.9301 | 0.8552 | 0.8508 | 0.9403 | 0.9211 | 0.9380 | 0.9172 | 0.9350 | 0.9176 |
| | Cont. | 0.9581 | 0.9418 | 0.9606 | 0.9427 | 0.9481 | 0.9240 | 0.9576 | 0.9470 | 0.9575 | 0.9463 | 0.9571 | 0.9460 |
| | Stead. | 0.8567 | 0.8267 | 0.8350 | 0.7850 | 0.7398 | 0.7023 | 0.8484 | 0.8063 | 0.8475 | 0.8016 | 0.8405 | 0.8020 |
| | Cohes. | 0.6795 | 0.6837 | 0.6488 | 0.6509 | 0.6870 | 0.6828 | 0.6620 | 0.6834 | 0.6557 | 0.6676 | 0.6564 | 0.6653 |

