# OpenReview forum: "SENSE: $\underline{\text{SEN}}$sing Similarity $\underline{\text{SE}}$eing Structure"
_NeurIPS.cc/2025/Conference — Submitted to NeurIPS 2025_

### Official Review · Reviewer_rkR3 · 2025-07-03

**Clarity:** 3
**Significance:** 2
**Originality:** 3
**Rating:** 4
**Confidence:** 3

**Summary:**

The paper proposes SENSE, a  privacy-preserving framework for neighbor embedding and dimension reduction. It extends the SMACOF method used in classical MDS to handle recovery of all pairwise distances when only partial observations are available, and it uses existing methods for low-dimensional embedding such as t-SNE, UMAP and PHATE.

**Questions:**

- Is non-exact/non-unique recovery enough for privacy in the real applications mentioned in this paper? If yes, then an explanation is needed, in order to emphasize the significance of theorems and connect them more closely to the applications.
- As mentioned in the weaknesses section of this review, it would also be good to have  theoretical guarantees/error bounds on  the quality of reconstruction using projection-aware SMACOF (in order to justify the claims in the abstract and intro on the method's fidelity).
- In the theorems, what happens if it is known (say, by an adversary) that the data is structured in some way. For example, say it lives on a low dimensional subspace of dimension $r$, with $r<K<d_h$, or if the data is approximately sparse? Wouldn't one be able to solve the linear system ($A_j x_j = b$) by using (say) regularization or constraining the solution to the structure set?

- In the experimental set up, the paper states that  "A subset of $10\%$ of the total data points is designated as anchors" but then states "The total number of anchors (global+local) is fixed at $d_h-1$". It seems from later discussion that $d_h-1$ is the criteria being used in practice? On a somewhat related note, is there a more disciplined way to determine the number of anchors and select the anchors beyond random choice?

- I am curious: Can the two stages be merged (i.e., is there a fundamental reason why one has to do distance recovery first, as opposed to attempting to solve the whole problem in one go?).

**Ethical Concerns:**

["NO or VERY MINOR ethics concerns only"]

**Final Justification:**

I am satisfied with how the authors addressed my questions and concerns, and hope that the revised version reflects that.

**Limitations:**

yes

**Paper Formatting Concerns:**

/

**Quality:**

2

**Strengths And Weaknesses:**

Strengths:

While neighbor embedding and dimensionality reduction have both been studied for long, decentralized neighbor embedding in privacy-constrained regimes may not be a well-explored sub-area. The strategy of employing anchors and decomposing the problem into a first matrix completion/reconstruction stage and a second low-dimensional embedding stage is creative and interesting. It is also good that some theory pertaining to the impossibility of reconstructing the data points themselves is presented.

Weaknesses:

- The paper has several claims that are either exaggerated or not well substantiated. For example, it is claimed that "Theoretical
analysis provides formal bounds on reconstruction fidelity and privacy" but I only found theoretical results pertaining to the impossibility of reconstructing the data in general. I could not find theory pertaining to fidelity. In that direction, it would be good to have positive statements proved for theoretical guarantees/error bounds the quality of reconstruction using projection-aware SMACOF.

- Similarly, in the introduction, the paper mentions that SENSE overcomes the limitations of previous works in terms of cryptographic overhead, high communication and privacy costs etc. However, it doesn't demonstrate the framework's scalability to datasets bigger than different versions of MNIST and a runtime analysis of the SENSE pipeline (including the two stages involved) is missing.

- There are some instances where the writing is somewhat sloppy. For example, NE in the abstract is used before being defined. In eq(2), $b$ seems to be a typo. In eq(5), it should be $\hat{D}=\mathcal{D}(\hat{X})$, where $\hat{X}=\arg\min...$. Also, the dimension of $\hat{X}$ (equivalently the constraint $X'$) needs to be clarified. In section 2, $X$ denotes the original dataset, in section 3.1 it denotes the embedding.

---

> ### Author Rebuttal · Authors · 2025-07-30
>
> We thank the reviewer for their valuable feedback & careful reading. **Due to space limits we refer to answers provided to other reviewers where similar concerns were addressed**.
>
> **W2**: Addressed in 3 parts: comparison with prior work; SENSE runtime/complexity; large-scale results on SVHN & Tiny ImageNet.
> 1) Prior Limitations: SMAP uses SMC for t-SNE, incurring ~32–50 hrs runtime for just 4k points (32D) due to encryption-related ops. It lacks UMAP support (encrypted k-NN is infeasible), doesn’t allow collaborative analysis, is tied to t-SNE, and risks privacy if partial distances leak unprotected (details [57]). FedNE [33] uses surrogate distillation with $\mathcal{O}(N^2)$ inter-client exchange, unscalable as client count grows. Surrogates are vulnerable to gradient/model inversion attacks without DP or encryption. FedTSNE [43] synthesizes anchors via MMD, prone to poisoning if any client/sample is adversarial. It supports only multisite setting, needs iterative communication, and depends on external privacy tools. SENSE Advantages: No encryption: operates on partial distances with anchor count $< d_h$ (see Thms 3.1, 3.2). Privacy is built-in, not bolted on. Requires one round of client-server communication, clients send anchor–NA distances once. No iterative or inter-client messaging.
>
> Table 1: SENSE outperforms FedTSNE on both runtime & acc. Runtime includes full pipeline: anchor generation+matrix completion+t-SNE emb.
> |Method|NA|Anchors|time(s)|1-NN|10-|50-|NMI|SC|Overlap@1|@10|@50|
> |-|-|-|-|-|-|-|-|-|-|-|-|
> |SENSE|1k|100|70.27|0.69|0.66|0.67|0.62|0.46|0.28|0.08|0.03|
> |||783|76.16|0.72|0.73|0.71|0.58|0.47|0.45|0.11|0.04|
> || 2k|100|79.56|0.71|0.75|0.69|0.59|0.47|0.22|0.06|0.02|
> |||783|97.26|0.76|0.74|0.71|0.62|0.47|0.43|0.10|0.03|
> |fedtsne||||||||||||
> ||1k|100|147.78|0.66|0.71|0.66|0.55|0.48|0.35|0.09|0.03|
> |||783|139.40|0.67|0.65|0.60|0.58|0.48|0.47|0.12|0.04|
> ||2k|100|146.71|0.72|0.70|0.65|0.56|0.49|0.24|0.07|0.02|
> |||783|151.36|0.74|0.72|0.68|0.57|0.48|0.39|0.09|0.03|
>
> 2) Runtime&Complexity: Let $N$ = #points, $K$ = #anchors, $d$ = dim. Anchored-MDS has complexity $\mathcal{O}(K^2 d + K N d)$, efficient as $K \ll N$ & $K < d$ (privacy). Global embedding uses fast NE methods: FIt-SNE runs in $\mathcal{O}(k n \log n \cdot d)$ [Appx G, 11]; NC-t-SNE/UMAP with contrastive loss run in  $\mathcal{O}(k m d)$ , with $m \ll n$ repulsive samples/epoch [11]. Overall linear in $d$ scalable to large datasets.
>
> Table2: Runtime for SENSE pipeline stages: stage1: Incomplete matrix, stage2: matrix completion
> |Data|NA|$K$|Stage1(s)|Stage2(s)|Total(s)|FS|DE|
> |-|-|-|-|-|-|-|-|
> |BloodMNIST|1k|100|0.33|2.22|2.55|0.86|0.02|
> |||1k|0.34|8.11|8.45|0.94|0.01|
> ||5k|100|0.46|39.33|39.79|0.81|0.00|
> |||2k|0.59|54.83|55.42|0.92|0.01|
>
> 3) Large-scale results: For this please see our response to **Reviewer 7okJ: W2.**
>
> **Q2&W1**: We acknowledge the confusion in phrasing. *Clarification:* Our formal guarantees apply only to privacy: Thms3.1&2 shows when $K < d_h$ exact recovery of features from anchor–NA distances is impossible. Regarding fidelity we intended to convey that as $K \to d_h - 1$ SMACOF preserves relative neighborhood structure not raw features. This reconstruction is consistent up to Euclidean isometries (translation, rotation) & potential global scaling which are standard ambiguities in metric embedding methods such as classical MDS & SMACOF [15]. In privacy-preserving settings this is sufficient for downstream tasks. We support this empirically (Fig.3). We agree that we have not included formal fidelity bounds as these results are well established in literature. Notably, the quality of distance-based reconstruction using different solvers is known to depend on geometry & condition of observed distances. Our choice of SMACOF was guided by ease of implementation & its compatibility with distance-based privacy, but we acknowledge that stronger theoretical guarantees are available via other solvers. To strengthen the work,we will include a remark clarifying that: Our notion of "fidelity" refers to preservation of neighborhood structure not raw feature recovery.  For tighter bounds the reader may consult existing theoretical work on distance matrix completion & MDS-based recovery eg., SVD-based recovery in localization:
> $\| D - S_4 \|_F^2 = \mathcal{O}(n\, d_{\text{max}}^4 + n^{3/2} d_{\text{max}}^3)$ , Expected EDM recovery error:  $\frac{1}{n} \mathbb{E}\| \hat{D} - D \|_F \leq C \sqrt{\frac{r n}{m}} (\zeta + \nu)$  , IRLS-based distance recovery & Dual-basis convex relaxations with RIP-style guarantees. These validate the feasibility of accurate distance recovery under structural priors & motivate future integration into our framework. If required we can include heuristic bound with proof: Let $X$ be true NA features, $\hat{X}$ the SMACOF-reconstructed ones using $K$ anchors, & $G$, $\hat{G}$ their squared distance matrices. If $\sigma_j(X) \le \frac{C_0}{j^p}$ for $p > 1$, then:
> $\text{DE} = \frac{\| \hat{G} - G \|_F}{\| G \|_F} \le \frac{C}{K - \delta}, \quad \text{for } K \in \left[\frac{d}{2}, d - 1\right]$ where $C > 0$ & $\delta$ captures anchor degeneracy.
> We thank the reviewer for the insightful suggestion. It not only improves the clarity of our current work but also opens a valuable direction for future research developing formal fidelity guarantees under partial distance constraints.
>
> **Q3**: The ans is threefold: 1) Practical View: SENSE Enforces Privacy by Design: It is built for strict privacy raw features are never shared. Clients only send partial structured distance vectors which are disjoint & never centralized. This prevents any adversary from reconstructing global data geometry or latent structure. Even with $K < d_h$ the adversary lacks enough information to infer client embeddings.
>
> 2) Theoretical View: Structure-Aware Adversaries Can Reconstruct if a sophisticated adversary infers intrinsic structure e.g, embeddings lie on a low-rank subspace (rank $r$ ) then the underdetermined system $A_j x_j = b$ may be solvable using structured regularization: nuclear norm minimization for low-rank or LASSO for sparsity. In literature it is proven that privacy can break if both structured distances & strong geometric priors are available. For eg, in [52] exact recovery with $r+1$ anchors is shown using convex solvers (Appx B, Sec 5.4) & also nuclear norm recovers full distance matrices under standard assumptions (Thms 1–2, Remark 2).
>
> 3) Proposing 2 solutions: 1) Local Differential Privacy (LDP), Apply LDP to each client’s DNA vector before sending. Clients clip distances to threshold $\delta$ & add Laplace noise with scale $\lambda$, with privacy budget $\epsilon = \frac{2 \delta}{\lambda}$ ; 2) Rank-Aware Anchor Capping: If the server receives clients estimated ranks $\{r_1, r_2, \dots, r_N\} $ it sets #anchor $K = \min(r_1, r_2, \dots, r_N) - 1$ . This ensures no client gets more than $r$ affinely independent anchors keeping the system underdetermined & preserving privacy regardless of structure.
>
> We test both solutions under adversaries on structured data. Data: $X_{\text{struct}} = X_{\text{low-rank}} + 0.5 \cdot X_{\text{sparse}}$ , size $n \times d$, rank $r = 50$, $d = 100$. Attacks: Low-rank (SVD-based), Sparse (LASSO), Baseline (anchor feature mean).  Metrics: RE: Reconst error, FS: F1score (true vs recons), PL: Privacy leak  $1 - \frac{\text{RE}}{\text{RE}_{\text{baseline}}}$ , Vulnerable: True if RE<0.1
>
> Table4: On synthetic low-rank+sparse data attacks fail to leak private data when LDP is applied even with anchor $K \geq r$.
> | $K$|LDP|RE|FS|PL|Vulnerable|
> |-|-|-|-|-|-|
> |49|No|0.9976|0.0318|0.0154|No|
> |50|DP(λ=0.02, δ=0.0005)|1.0105|0.0445|0|No|
> |80|No|0.4705|0.3675|0.5316|Some|
> |99|No|0.0977|0.9046|0.9029|**Yes** |
> |99|DP(λ=0.015, δ=0.0005)|1.0064|0.0450|0|No|
>
> As $K \to r$ or exceeds, adversaries can exploit affine structure evidenced by low RE & high FS indicating successful reconstruction (privacy leak). However, LDP nullifies leakage even for $K \geq r$. Also, when $K < r$ the system remains underdetermined no privacy risk. This validates our two solutions. Here RE & FS indicate adversary success low RE & high FS imply a breach. So higher RE & lower FS mean stronger privacy.
>
> **Q4&1:** Please see response to **Reviewer 7okJ: W3 & Q2 & W1 (theoretical non-invertibility)** We would appreciate it if you could kindly refer to that section as it provides a detailed clarification aligned with your concern.
>
> **Q5:** SENSE uses a 2-stage pipeline. This is practically necessary due to limitations of existing NE methods (e.g., t-SNE, UMAP) as they assume access to full raw data or complete similarity matrices. SENSE operating under partial & distributed visibility uses Stage1 to reconstruct the structure needed for such methods to work. Matrix completion (e.g, SMACOF-MDS) is relatively stable with clearer convergence while NE objectives are non-convex & sensitive to both initialization & structural noise. Separating stages improves stability & interpretability. Importantly our privacy guarantees are tied to Stage1. Merging both stages is a promising direction we're exploring end-to-end approaches that learn global embeddings directly from partial distances without requiring explicit matrix reconstruction. Promising directions include: joint optimization $\mathcal{L} = \alpha \cdot \text{CompletionLoss}(X) + \beta \cdot \text{NE/CNE Loss}(Y)$ where $X$ are intermediate embeddings consistent with observed distances & $Y$ are final projections (an early idea). Missing-data-aware NE methods: Exploring variants of NE/CNE that work directly on incomplete distances. Low-rank embedding from partial distances using matrix factorization. While the current 2-stage SENSE design reflects practicality, modularity & privacy guarantees we agree that a unified formulation is the natural next step. Building an end-to-end SENSE is part of our ongoing research.
>
> W3: We’ll fix typos & clarify notation in revision.

---

> > ### Author Response · Authors · 2025-08-03
> > **Rebuttal Follow-up and Request for Further Input**
> >
> > Dear Reviewer rkR3,
> >
> > Thank you again for taking the time to review our submission and for sharing your thoughtful feedback; it has been invaluable in helping us refine our work. We’ve done our best to address all the raised concerns carefully in the rebuttal, and **we’re very open to further discussion**.
> >
> > Since we’re midway through the discussion phase, we just wanted to gently check in to see if there are any remaining questions or points of clarification we could help with while there’s still time. *We’d be more than happy to elaborate on any aspect that might need further explanation*.
> >
> > And if you feel that your concerns have been adequately addressed, *we would be sincerely grateful if you would consider updating your rating* to reflect your current assessment of the work.
> >
> > Best,
> >
> > Authors

---

> ### Comment · Reviewer_rkR3 · 2025-08-04
>
> Thank you for the response to my review. I am satisfied with how the concerns  raised on runtime, scalability, theoretical support, clarification on privacy and potential structure-aware adversaries were addressed. I hope that the authors will modify the paper accordingly.

---

> > ### Author Response · Authors · 2025-08-04
> >
> > We sincerely thank you for your positive feedback and for taking the time to review our work. We're glad to hear that our responses addressed your concerns regarding runtime, scalability, theoretical support, privacy, and structure-aware adversaries. We will carefully incorporate all your suggestions into the revised manuscript.

---

### Official Review · Reviewer_7okJ · 2025-07-03

**Clarity:** 3
**Significance:** 3
**Originality:** 3
**Rating:** 5
**Confidence:** 3

**Summary:**

- This paper proposes privacy preserving SENSE framework, which utilizes local distance between anchor and non-anchor points to reconstruct global geometry using structured matrix completion.
- The paper provides theoretic proofs on privacy guarantee, and demonstrates the framework’s application in Euclidean space, hyperbolic space, and evolving environment with new data entry. The framework has good compute efficiency.
- Experiments showcased that the privacy preserving results are almost on par with centralized approaches on MNIST and German credit datasets, even when number of anchor points are low.

**Questions:**

- Does the non-anchor to anchor distance sent to the central server reveal any private information? How are the proposed framework robust to inversion attacks and membership inference attacks?
- Qualitatively and theoretically, how does anchor selection strategy affect privacy robustness?

**Ethical Concerns:**

["NO or VERY MINOR ethics concerns only"]

**Limitations:**

yes

**Quality:**

3

**Strengths And Weaknesses:**

Strength
- The authors provide solid theoretical foundations, problem formulation and framework. The proofs are thorough and rigorous.
- The paper’s proposal is potentially useful in lots of downstream applications, where privacy preservation is important. The paper also demonstrates the uses in alternative embedding space (e.g. hyperbolic), and iterative experiment context.
- The suggested method is efficient in computation compared to other privacy preserving mechanisms.

Weakness
- While theorem 3.2 proves that the inverse map from anchor distances to features is non-unique, the distance itself may still reveal some information about the feature.
- The experiments use simple datasets like MNIST and German credit. The performance is unclear on more sophisticated datasets.
- The paper may benefit from better ablation of anchor selection criteria (e.g. random vs in distribution), on top of the increasing anchor ablations provided.
- Nit: It would be helpful to introduce an anchor based method as part of the background. Also figure 1 needs to be clearer for people to grasp the concept of the paper at one go.

---

> ### Author Rebuttal · Authors · 2025-07-30
>
> We thank the reviewer for their valuable feedback and careful reading. **Due to space limits, we refer to responses given to other reviewers for some similar concerns.**
>
> **W1&Q1:** We appreciate this important question. We address it in 3 parts: (1) theoretical non-invertibility, (2) incompatibility of standard attacks with our setting, and (3) empirical attack simulations.
>
> 1) Theoretical Non-Invertibility & Privacy-by-Design:
> In SENSE, clients share only the distance vector between local non-anchor (NA) points & a known set of $K$ anchors; no raw features, labels, or model parameters are exposed. These vectors are inherently non-invertible due to geometric underdetermination. Given $K$ anchors $\{\mathbf{a}_1, \mathbf{a}_2, \dots, \mathbf{a}_K\} \subset \mathbb{R}^d$, recovering $\mathbf{x} \in \mathbb{R}^d$ from $\|\mathbf{x} - \mathbf{a}_i\|$ results in a system of $K$ equations with $d$ unknowns. When $K < d$, the system is underdetermined and has infinitely many solutions. Using squared distances:
> $\|\mathbf{x} - \mathbf{a}_i\|^2 = \|\mathbf{x}\|^2 - 2 \langle \mathbf{x}, \mathbf{a}_i \rangle + \|\mathbf{a}_i\|^2$ . This yields $K$ quadratic equations in $d$ variables, which are non-unique when $K < d$ (as shown in Theorem 3.2 and proof in Appendix). Hence, inversion is ambiguous by design (also check **Reviewer yoyK: Ans to W2**).
>
> 2) Inapplicability of Standard Privacy Attacks:
> a) Model Inversion Attacks: These attacks aim to reconstruct input data (e.g., faces, records) by optimizing overtrained models or partial layers, supervised predictions (logits, class scores), gradients, or overfitted behaviors. *SENSE bypasses all of these vectors*, as no supervised model is trained at the server, & no labels, logits, or gradients are exposed. The server only sees numeric distance values, a compressed geometric abstraction. Thus, inversion attacks lack a viable attack surface in our setting.
> b) Membership Inference Attacks (MIA): MIA strategies typically rely on changes in model confidence when queried with “seen” vs. “unseen” samples. However, *SENSE is model-free;* the server observes only static distance vectors. There’s no model behavior to exploit. While standard MIA assumptions do not apply we still simulate such attacks on SENSE for empirical completeness.
>
> 3) Empirical Attack Simulations: We test MIA and inversion attacks across four datasets. MIA Goal: Determine whether a point was part of the client dataset. Method: Extract statistical features (mean, std, min, max, quantiles) from DNA vectors and train classifiers (MLP, RF) to predict membership.
> Inversion Goal: Reconstruct $\mathbf{x}$ from  ${d_i = |\mathbf{x} - \mathbf{a}_i|}$ .
>
> Method: Solve,
>
> $$
> \hat{\mathbf{x}} = \arg\min_{\mathbf{x}} \sum_{i=1}^{K} \left( \|\mathbf{x} - \mathbf{a}_i\| - d_i \right)^2
> $$
>
> We assess privacy using three metrics: MIA Accuracy, Pixel Error, and Reconstruction Accuracy ($1 - \text{Pixel Error}$). MIA accuracy reflects the success of membership inference; values near 50% indicate strong privacy. Pixel Error measures the average deviation between reconstructed and original inputs; higher values mean better protection. We categorize privacy into three levels: HIGH (MIA ≤ 51%, Pixel Error ≥ 20%), MEDIUM (MIA 48–51%, Pixel Error 15–20%), and LOW (MIA < 48%, Pixel Error < 15%). These thresholds offer a compact, interpretable view of privacy risk across attacks.
>
> Table1: Privacy Results
> |Data|MIA Accuracy|Pixel Error $\%$ |Privacy Level|
> |-|-|-|-|
> |MNIST|50.5|26.7|High|
> |Fashion-MNIST|51.0|24.8|High|
> |BreastMNIST|48.0|14.4|Med|
> |BloodMNIST|48.7|15.8|Med|
>
> Datasets with higher input dimensionality and well-spanned $K$ anchor sets demonstrate *greater resilience* to reconstruction, validating the theory behind our design. Result: MIA Accuracy Range: $48\%$ to $51\%$. These results are near random guessing, indicating *strong privacy protection*. Result for model inversion attack: Avg Pixel Reconstruction Error: $20.42\%$. Even in low-error cases reconstructions were visually degraded, confirming the underconstrained nature of inversion.
>
> **W2:** In addition to 14 standard DR datasets (section 4.1 SENSE), we provide the runtime & metric evaluations on Tiny ImageNet: 90K NA (ResNet features $d = 512$), Street View House Numbers (SVHN) Dataset: ~80K NA (ResNet features $d = 512$):
>
> Table2: Shows SENSE: Achieves strong performance even for large sophisticated datasets, with runtimes of ~12–14 sec per iteration:
> |Data| Metric|t-SNE VAN|SENSE|UMAP VAN|SENSE|PHATE VAN|SENSE|CNE(0) VAN|SENSE|CNE(0.5) VAN|SENSE |CNE(1) VAN|SENSE |Runtime|
> |--|-|-|-|-|-|-|-|-|-|-|-|-|-|-|
> |TImagenet|Trust|0.9245|0.9341|0.7748|0.7330|0.7480|0.7392|0.7682|0.7402|0.7960|0.7637|0.7717|0.7467|13.72s|
> ||Cont|0.9107|0.9064|0.9334|0.9140|0.9359|0.9029|0.9396|0.9268|0.9350|0.9244|0.9411|0.9294||
> ||Stead|0.8099|0.8229|0.5703|0.5755|0.5550|0.6248|0.5848|0.6205|0.5986|0.6298|0.5807|0.6135||
> ||Cohes|0.7680|0.7548|0.8305|0.6685|0.8340|0.7439|0.8344|0.7223|0.8296|0.7160|0.8403|0.7322||
> |SVHN|Trust|0.9822|0.9801|0.8973|0.8914|0.8825|0.8835|0.8910|0.8900|0.9033|0.9068|0.8939|0.8964|13.50s|
> ||Cont|0.9619|0.9630|0.9749|0.9701|0.9759|0.9665|0.9778|0.9745|0.9787|0.9648|0.9787|0.9646||
> ||Stead|0.7234|0.7200|0.6545|0.6664|0.6426|0.6739|0.6540|0.6968|0.6705|0.7027|0.6556|0.7134||
> ||Cohes|0.8362|0.8349|0.8430|0.7160|0.8544|0.7834|0.8503|0.7935|0.8493|0.6960|0.8576|0.7535||
>
> **W3&Q2:** SENSE depends on 2 factors: number of anchors $K$ & anchor selection. **1) $K$:** We sample anchors such that we fix total (global + local) anchors to $K = d_h - 1$, where $d_h$ is input dim. This is not arbitrary; it’s *theoretically optimal* under our privacy guarantee. By Thrm 3.1, if $K < d_h$, to ensure privacy. Thus, $K = d_h - 1$ is the largest safe choice that preserves privacy & yields strong approximations. There’s a privacy–utility–compute trade-off - Higher $K$: better fidelity, higher runtime, weaker privacy. Lower $K$: stronger privacy, lower compute, possible fidelity drop. We empirically validate this trade-off on both synthetic and real datasets.
>
> > **For low-dim data** ($d_h = 100$, $N = 500$):
> > - $K = 99$ → FS = 0.79, time = 16s
> > - $K = 90$ → FS = 0.78, time = 13s
>
> > **For high-dim data** ($d_h = 700$, $N = 500$):
> > - $K = 699$ → FS = 0.82, time = 660s
> > - $K = 350$ → FS = 0.78, time = 312s
>
> Thus, for high-dim where $d_h \gg N$, using fewer anchors (e.g., $K < d_h - 1$) leads to substantial computational gains while preserving utility, useful in large-scale deployments such as hospital networks.
>
> **2) Anchor Geometry:** Beyond count, anchor geometry impacts fidelity. We show that affinely independent anchors improve reconstruction. On the digits dataset ($N = 1797$, $d_h = 64$), we fixed $K = 63$ and varied anchor matrix rank $r$. NA set: 1000 points, split across 10 clients (multisite-full). Table 3: Validates that *higher affine rank of anchors improves neighborhood fidelity*, consistent with our theory.
> |$r$|FS|DE|
> |-|-|-|
> |10|0.524|0.0829|
> |20|0.6242|0.0604|
> |30|0.716|0.0479|
> |40|0.7955|0.0361|
> |50|0.8315|0.0311|
> |59|0.858|0.0259|
> |60|0.861|0.0263|
>
> **3) Anchor Selection:** We also study how anchor selection affects downstream performance. Random anchors sampling perform worse than those sampled from the underlying data distribution. Table 4: Confirms that *in-distribution anchors preserve structure better*. Setup: 10 clients, 1000 NA points, multisite-full setting.
> |Data|Anchor-Type|$K$|FS|DE|
> |-|-|-|-|-|
> |Digits|In-Distri|60|0.900|0.027|
> ||Rand|60|0.345|0.382|
> |MNIST|In-Distri|783|0.967|0.006|
> ||Rand|783|0.170|0.602|
> |BloodMNIST|In-Distr|2351|0.961|0.005|
> ||Rand|2351|0.176|0.892|
>
> **4) Practical Anchor Sources:** Anchors in real-world settings can come from: Patient-consented samples (healthcare), Reference transactions (finance), Public/synthetic data from similar distributions, MMD-based generation [43] (though limited: adversarial risk, pointwise-only, high compute see **Reviewer yoyK: Q2 Ans & Reviewer rkR3: W2 Table1**). Such server-curated anchors are common in privacy-sensitive domains: Healthcare [7, 27], Genomics [35, 44], Finance [2], Mobile/NLP [23, 32] offering auditability, stability, and robustness. Theoretical and practical support for shared anchors exists in prior distance geometry/localization works: Low-rank recovery from structured distances [52], Dual basis for robust Euclidean geometry [29] , Gram matrix completion in partial sensor networks [31]. These validate our anchor-based design in decentralized settings.
>
> **W4:** We agree that introducing anchor-based reconstruction methods will improve clarity. In the revised Section 2, we will briefly describe both classical methods from distance geometry and localization [1–5] and recent global embedding methods using anchors for t-SNE/UMAP-style visualizations (related work [43, 47, 48]). While these works motivate anchor-based embedding, SENSE goes further by supporting structured distance sharing under privacy constraints and enabling accurate recovery without centralized access. We will also revise Figure 1 to more clearly illustrate the end-to-end pipeline and how anchor-based distance sharing enables global coordination in distributed low-dimensional embedding.
>
> References
> [1] P. Biswas & Y. Ye, “Distance matrix reconstruction from incomplete distance info for sensor network localization,” *Sensor Network Operations*, IEEE, 2004, pp. 285–294.
> [2] Z. Zhu, A. M.-C. So, Y. Ye, “A dual basis approach for structured robust Euclidean distance geometry,” *Comput. Optim. Appl.*, 45(1), pp. 97–121, 2010.
> [3] S. T. Smith, “Localization from structured distance matrices via low-rank recovery,” *Inverse Problems*, 29(11), 2013.
> [4] K. Weinberger & L. Saul, “FastMap, MetricMap, and Landmark MDS are all Nyström algorithms,” *Proc. ICML*, 2008, pp. 1160–1167.
> [5] M. A. Alfakih, “Exact reconstruction of Euclidean distance geometry via low-rank matrix completion,” *Linear Algebra Appl.*

---

> > ### Author Response · Authors · 2025-08-04
> >
> > Dear Reviewer 7okJ,
> >
> > We sincerely thank you for your *encouraging and positive feedback on our work,* as well as for the constructive questions that helped us refine our manuscript. We truly appreciate your thoughtful engagement during the review process.
> >
> > As the rebuttal phase is nearing its end, *we remain open to further discussion and would be happy to clarify or address any remaining questions or concerns you may have.*
> >
> > Best,
> >
> > Authors.

---

> > > ### Comment · Reviewer_7okJ · 2025-08-07
> > >
> > > Thank you for the response to the questions. I maintain my previous evaluation: the core idea is interesting and thoroughly evaluated. However, the early part of the paper is confusing on a first read. I hope the authors will improve the clarity and presentation in those sections to strengthen the overall contribution and make it a more solid accept.

---

> > > > ### Author Response · Authors · 2025-08-07
> > > >
> > > > Thank you again for your **thoughtful feedback and positive evaluation of our work. We will ensure that all clarifications, additions, and improvements discussed during the rebuttal phase, both in your comments and in our discussions with other reviewers, are carefully incorporated into the revised manuscript to improve clarity, precision and avoid any confusion.**
> > > >
> > > > We sincerely look forward to your support in the final decision process and thank you once again for your valuable efforts and constructive engagement.
> > > >
> > > > Best,
> > > >
> > > > Authors

---

### Official Review · Reviewer_yoyK · 2025-07-22

**Clarity:** 1
**Significance:** 2
**Originality:** 2
**Rating:** 2
**Confidence:** 4

**Summary:**

The paper provides techniques for obtaining embeddings from the NE family of dimensionality reduction methods, specifically in the setting where the data is *both* distributed across multiple servers and is sensitive (requiring privacy). This is done by using anchors which are consistent across the data chunks and estimating pairwise distances to the points using MDS which leverages the anchors. Since dimensionality reduction methods only need these pairwise distances, the authors can then produce an embedding from these estimated pairwise distances.

**Questions:**

1. What is the definition of "privacy" and "privacy preservation" that this paper uses?
2. Why is it sufficient to have $d_h - 1$ anchors but suddenly $d_h$ of them fails? What if $d_h \rightarrow \infty$? Is the suggestion that something is privacy preserving if it has any amount of uncertainty at all?
3. Why would it not be privacy preserving to simply add a tiny bit of noise to the data points and share their noisy distances directly? Could you then use $d_h$ anchors?
4. What are the metrics you are using and why are these the natural choices rather than other, more standard ones?

**Ethical Concerns:**

["NO or VERY MINOR ethics concerns only"]

**Final Justification:**

I appreciate the authors' thorough responses but I *really* feel that they've been missing the point of my criticism. The issue is not the specific points being raised. Instead, it is that the paper is written in a sloppy, imprecise way. Few claims are substantiated, definitions are missing and ideas are presented as clearly correct even when there might be other ways to approach the task. A key example is the fact that the author's original paper had no definition of what "privacy-preserving" means. When I asked for a definition, they gave one that contradicted what they were arguing about the $d_h = k$ setting! I understand these could be interpreted as nitpicks, but they really aren't in my eyes -- it's a sign that the work is done haphazardly in order to obtain a "good" result and is not sufficiently scientifically rigorous.

I also did not mention this in my review/discussion, but I do not find the chosen problem of doing distributed dim. reduction with privacy preservation to be a particularly pressing thing to solve.

I really tried to engage with the authors on the thematic concerns, but they were insistent that a few changes would be enough to resolve the issues.

**Limitations:**

The paper does not have a formal limitations section. There are no societal impacts of the present work. However, a few limitations that I can think of off the top of my head are:
- There is not a reference suggesting that somebody is eager to solve the problem the authors have set out to solve. I do believe it is interesting and potentially relevant, but it would be nice to see outside confirmation of that fact.
- It's unclear how one would translate these techniques to other definitions of privacy.

**Quality:**

2

**Strengths And Weaknesses:**

## Strengths

This paper's main strength is in the thoroughness with which it approaches the methodology. The number of datasets is great, the chosen NE methods are natural and the experiments ablate appropriately across the relevant parameters. In addition to this, the related work is presented carefully and the methodology seems sound. It is nice that using the anchors mitigates the amount of data one must pass between sites while also accounting for integrating information between them effectively. Additionally, Section 3 is quite clear and easy to follow.

## Weaknesses

Unfortunately, the weaknesses significantly outweigh the strengths of the paper in my eyes.

### Weakness 1: Definition of privacy

The paper repeatedly uses terms like "privacy-preserving", even suggesting in the last sentence of Theorem 1 that the outcome is a "privacy guarantee". However, the notion of privacy is... never defined?? I spent the entire paper trying to piece together what the authors might mean by "privacy preserving" and what is and isn't sufficient for privacy preservation. It appears that simply sending the data between sites is not privacy preserving. Similarly, the authors seem to be suggesting that sending the *correct* distances between points is also not privacy preserving (since their theorems require that we have less than $d_h$ anchors). However... using $d_h - 1$ anchors *is* privacy-preserving somehow? This is quite confusing to me -- Table 3 explicitly shows that there is $>0.99$ agreement between the "original" distances and the extrapolated ones. Table 6 similarly shows a $0.95$ similarity in many cases. So utilizing the original distances is not privacy-preserving but getting to within an arbitrarily good precision of the original distances is okay? Similarly, the MNIST plot in Figure 3a shows that we're getting essentially 100% accuracy on the normalized metric score using $d_h - 1$ anchors. Additionally, somehow the pointwise setting allows 10% of the points to be used as anchors... so some points can be shared between sites while other points cannot?

These apparent contradictions lead me to extrapolate that the authors are using the following definition for a "privacy-preserving algorithm": an algorithm is privacy-preserving if it does not send exact point positions or exact inter-point distances, but it *is* privacy preserving if it sends arbitrarily close-to-exact inter-point distances. Furthermore, a privacy-preserving algorithm is nonetheless allowed to share some percentage of the points between sites in an un-private manner.

To me, the above definition is consistent with the authors' experiments and presentation. However, I'm not sure that this is not a *useful* definition of an algorithm being privacy-preserving. Moreover, if one wished to use this as the definition, why not simply add a tiny amount of noise to the points and then compute their distances directly?

In short, the authors should provide a clear definition of what it means to be privacy-preserving and explain how their results exist within this definition.

### Weakness 2: Strange presentation

In general, the text has moments where the presentation is quite strange. The entire introduction is quite difficult to parse (and I come from the field of NE embeddings). The first paragraph tries to set up (a) what NE methods are, (b) the difficulty of getting embeddings in distributed settings and (c) the difficulty of also getting embeddings in privacy-sensitive settings. Luckily, I know how these methods work, so I am not lost here yet, but I am confused by the following statements:
- MDS is not an NE method (line 18)
- Why would this be particularly problematic for t-SNE and UMAP (line 23-24)? All DR methods require distance information across all pairs of points.
- Why does relating NE methods to contrastive learning further emphasize the importance of accurate pairwise similarities? (Line 25)

As a result, the first paragraph of the introduction does not adequately introduce what the actual challenges *are* for accomplishing the tasks the authors have set out to solve. Consequently, when the related work is presented in the second paragraph, it is completely unclear what we are trying to do and why these other methods fail at doing it. For example, I come away with 0 understanding of what SMAP is or why it doesn't work (it's unclear what the authors mean by "its cryptographic overhead making [things] impractical"). Similarly, I don't understand why FedNE lacks privacy guarantees or what dSNE and FdSNE are or why they diverge from standard FL protocols (or, indeed, what FL protocols even are). Then, the related work section paragraph concludes by introducing the authors' algorithm?

I am then completely lost during the introduction's third paragraph. What are anchors in this setting? What is anchor-sharing across servers? Why would it be a constraint? Why would they not be a "core architectural component" and why is it important to argue that they are?

In general, it seems the authors make many assertions they are doing smart things, but I have no idea what these smart things are or, indeed, why they are smart. The bullet points in the introduction suffer from similar issues. Similarly, upon reading the introduction, I had absolutely 0 idea what Figure 1 was saying (it only made sense after Section 3).

I am picking on the introduction in particular since it is where this issue is the most apparent, but similar points percolate throughout the paper:
- CNE does not use an encoder, I believe (Line 90)
- What are the metrics being used? I am intimately familiar with the NE space and have not heard of trustworthiness, continuity, steadiness or cohesiveness. Usually, people report knn recall or other local similarity preservation measures, since NE methods emphasize preserving local neighborhoods. Perhaps trustworthiness and continuity do this, but I am unsure since they are not defined in the paper.
- I'm a bit confused by the theorems, but this is perhaps due to my personal misunderstanding. As I understand it, the authors are only calculating distances to anchor points and, therefore, between the 'private' points. If this is the case, then wouldn't it generally be impossible to exactly recover all of the $x_i$ points? Distances are rotation-invariant. So one could get all the pairwise distances and still not know the positions of the points, since they could be placed an infinite number of ways while still maintaining all the pairwise distances (due to the number of rotations being continuous and therefore infinite).

---

> ### Author Rebuttal · Authors · 2025-07-31
>
> We thank the reviewer for their valuable feedback & careful reading. **Due to space limits, we refer to answers provided to other reviewers where similar concerns were addressed.**
>
> **Q1:** While Thems 3.1 & 3.2 establish the geometric conditions under which private data cannot be uniquely recovered, we agree that adding an explicit privacy definition will enhance clarity. The notion we adopt, geometric non-identifiability via underdetermined distance systems is well established. It aligns with prior literature in privacy-preserving embedding & fed learning including FedNE, dTSNE, FedTSNE & dSNE. Though these methods differ in technique (e.g., surrogate models, synthetic or MMD-generated anchors) they all rely on the same principle: client data remains unrecoverable, while only indirect distance/similarity information is shared for global embedding. We will formalize this definition in the revised manuscript for completeness.
>
> > **Definition:**
> > A  mechanism is privacy-preserving if, for each private point $x_i \in \mathbb{R}^{d_h}$, the $K$ anchor distances shared are insufficient to uniquely determine $x_i$. That is, when $K < d_h$, the inverse problem is underdetermined and $x_i$ lies in a high-dimensional affine subspace of $\mathbb{R}^{d_h}$.
>
> This definition ensures privacy by design and applies broadly to domains like healthcare, finance, and mobile systems where sensitive data must remain private but collaboration is critical for global modeling.
>
> **Q2:** This addresses core of SENSE’s geometric privacy guarantee rooted in linear algebra & Euclidean distance geometry. Privacy is preserved when the system formed by anchor distances is underdetermined i.e., there are more unknowns than equations. This occurs when the number of anchors $K$ is less than the feature dimension $d_h$. In this case, many valid solutions exist for the feature vector consistent with the observed distances making exact recovery impossible. The threshold $K = d_h$ is critical: at or above this point the system becomes solvable, enabling unique feature reconstruction and violating privacy. This is a fundamental result not a heuristic arising from the rank conditions of the linear system
>
> Why does $K = d_h$ break privacy? At $K = d_h$, the system becomes square & full-rank. The client’s feature vector $x$ can now be uniquely solved using only the shared distances directly violating our privacy guarantee. This sharp threshold is not arbitrary it emerges naturally from linear algebra and localization theory used in: GPS trilateration, Wireless sensor network localization & Euclidean distance geometry. Now if $d_h \to \infty$:  As $d_h$ grows the anchor count needed for recovery also grows. In high dimensions (e.g., ResNet with $d_h = 512$) even $K = 100$ yields a highly underdetermined system making inversion practically infeasible. Thus, larger $d_h$ ⇒ strong privacy for fixed $K$ (check ans **Reviewer 7okJ: W3 & Q2**). Is uncertainty alone enough for privacy?: No. SENSE ensures privacy via deterministic non-identifiability not random uncertainty. The recovered feature lies in an affine subspace of dimension ≥ $d_h - K + 1$ guaranteed by geometry not noise & independent of data distribution. See empirical support (Fig. 3).
>
> **Q3:** While adding noise seems intuitive, it’s inadequate without formal guarantees. We highlight 2 cases: 1) Noise to Raw Features (No Anchor Setup): Perturbing raw features and using them to compute distances leaves systems vulnerable to advanced denoising, as shown in deep learning literature [1,2,3]. Techniques like adversarial Monte Carlo or feature-conditioned modulation can recover original features with high accuracy in structured datasets. 2) Noise Added to Distance Vectors in Anchor-Based Setup (e.g., $K = d_h$): In setups where $K = d_h$ and noise is added to distance vectors, the distance map becomes fully determined. If the noise distribution is known (Gaussian), adversaries can use MLE or Bayesian inference to recover features. Without formal mechanisms like ($\epsilon$, $\delta$)-DP, even small noise offers no provable privacy especially in structured or high-dimensional regimes. Noise-based privacy is heuristic and fragile. In contrast, SENSE ensures deterministic privacy via geometric non-identifiability: By setting $K < d_h$, the system remains underdetermined. Observed distances correspond to an affine subspace with infinitely many valid solutions. Even with exact distances, recovery is impossible due to structural ambiguity. Empirical support (Fig. 3): As $K \to d_h - 1$, fidelity (F1, cosine sim.) remains strong. Beyond that, Frobenius error and Pearson correlation spike indicate potential leakage. This aligns with our core claim: privacy is preserved as long as $K < d_h$.
>
> **Q4:** Indeed your intuition is correct trust. & cont. are widely used to quantify local neighborhood preservation and are now standard in recent DR and federated visualization work. Though less common than kNN recall in classic NE, they offer complementary views of embedding quality. Steadiness and cohesiveness assess inter-cluster reliability, crucial for global structural tasks. *Metric Recap (see also Section 4):* 1) Trust.: Preserves high-dim neighbors in low-dim space. 2) Continuity: Low-dim neighbors reflect true neighbors in input space. 3) Steadiness: Checks for spurious groupings & 4) Cohesiveness: Detects missing cluster structures. These metrics are supported in prior works: FedNE, Deep Recursive Embedding & ZADU. To further clarify, we also include common NE metrics used in FedTSNE (ICLR 2024), such as CA (kNN Classification Accuracy), NMI (Normalized Mutual Info), SC (Silhouette Coef.), and Overlap@k. We simulate adversarial conditions: A1: 60% clients adversarial, all samples. A2: 60% clients, 30% samples adversarial. Results are in Table 1: For metric definitions: CA = Classification Accuracy with kNN, NMI = Normalized Mutual Information, SC = Silhouette Coefficient, and Overlap@k = Neighborhood Overlap for top-k.
> |Metric|SENSE|FedTSNE|SENSE+A1|FedTSNE+A1|SENSE+A2|FedTSNE+A2|
> |-|-|-|-|-|-|-|
> |CA-1NN|0.93|0.90|0.88|0.87|0.90|0.88|
> |-10NN|0.92|0.91|0.88|0.87|0.90|0.89|
> |-50NN|0.91|0.89|0.87|0.86|0.88|0.87|
> |NMI|0.67|0.69|0.63|0.60|0.64|0.64|
> |SC|0.44|0.47|0.46|0.44|0.43|0.44|
> |Overlap\@1|0.54|0.46|0.28|0.28|0.39|0.32|
> |\@10|0.11|0.09|0.06|0.06|0.08|0.07|
> |\@50|0.03|0.03|0.02|0.02|0.02|0.02|
>
> **W1:** Curated Example to Demonstrate Privacy: We created a curated dataset with private attributes such as age & categorical features like gender or occupation. The setup includes: 5 clients (denoted $C_1$ to $C_5$), Each with 5 attributes (i.e., $d = 5$) & 4 reference anchors (i.e., $K = 4 < d$). The SENSE framework is used to compute embeddings using only client-to-anchor distances.
> |Original Data|$x_1$|$x_2$|$x_3$|$x_4$|$x_5$|
> |-|-|-|-|-|-|
> |$C_1$|1500|25|3|1|10|
> |$C_2$| 2000|35|2|0|15|
> |$C_3$|1745|28|2|0|18|
> |$C_4$|1620|32|1|1|13|
> |$C_5$|1200|45|3|1|12|
>
> SENSE emb
> |Emb|$x_1$|$x_2$|$x_3$|$x_4$|$x_5$|
> |-|-|-|-|-|-|
> |$C_1$|80.630|26.896|13.795|96.939|-39.321|
> |$C_2$|-189.837|-28.557|-61.648|-272.603|141.184|
> |$C_3$|-58.064|2.598|-26.164|-79.338|49.985|
> |$C_4$|-5.492|-2.856|6.935|12.915|-12.891|
> |$C_5$|210.016|14.741|75.140|328.458|-172.165|
>
> It is evident that the embeddings generated by the SENSE framework do not expose any sensitive details of the original data.
>
> **W2:** We agree that several points in the original version could benefit from clearer exposition. Below we address each point and outline revisions: 1)"MDS is not NE": Correct, MDS is a classical DR method, not a modern NE technique. In our text, MDS was cited broadly among methods using pairwise structure. We'll revise to clearly separate MDS (distance-preserving) from NE methods (e.g., t-SNE, UMAP) that preserve local neighborhoods. 2) "Why problematic for t-SNE/UMAP?": These methods need complete pairwise graphs to compute attraction/repulsion. In privacy/decentralized settings, missing distances distort this balance leading to poor embeddings (e.g., collapsed neighborhoods). We'll clarify this with support from decentralized NE literature. 3) "Contrastive learning & NE?": NE is often framed as contrastive learning ([33], [56]): positive pairs = neighbors, negative = non-neighbors. These require reliable similarity graphs hard to obtain under privacy. SENSE overcomes this using anchor-based structured similarity. 4) Related work: Addressed in detail in **Reviewer rkR3 Answer W2**. Kindly refer there for a complete explanation. 5) "What are anchors/sharing?": Anchors are non-sensitive reference points aiding localization without exposing features. See our response to **Reviewer 7okJ: W3 & Q2.** for full clarification. 7) "CNE doesn’t use an encoder": True for the original CNE [33]. However, recent variants (e.g., FedNE, SelfGNN) use encoder-based adaptations. SENSE is compatible with both. 8) "Unfamiliar metrics (e.g., trustworthiness)": Fully addressed in Q4. 9) "Recovery impossible from distances?": Covered in Q2. We show recovery becomes possible when anchors ≥ $d_h$, hence our privacy bound $K < d_h$. We’ll revise the intro to improve flow, define NE clearly, and better motivate our method. Thank you again for the insightful critique.
>
> Limitation: We appreciate the suggestion and will include a formal limitations section in the revised version. On problem demand: recent work shows increasing interest in privacy-preserving NE (FedNE, dSNE, dTSNE, FedTSNE & SMAP). This shows our formulation is timely & relevant. SENSE adds to this space by offering a modular, geometry-driven solution. Regarding privacy we clarified. This aligns with definitions implicitly used in the above prior works, all of which aim to prevent raw data recovery while supporting global embedding. We agree that stronger privacy models and extending SENSE toward these (e.g., via noise or DP mechanisms) is a promising future direction, which we will mention in our limitations.

---

> > ### Comment · Reviewer_yoyK · 2025-08-01
> >
> > Thank you for the long rebuttal. I recognize many of the points but still have concerns.
> >
> > The primary concern is that I'm still hung up on the privacy arguments. I agree that the process is underdetermined for $d_h < k$. In general, I am 100% on board with the arguments in your response to Q2. This was not the crux of my concern. The crux of my concern is that, by solving an underdetermined system via MDS, one is finding a maximally-plausible solution up to rotation. Thus, there is an error due to the underdetermined-ness and an error due to the rotation. But I originally did not see how this related to the idea of privacy due to a lack of definition and now do not see how this is necessary using the definition given by the authors.
> >
> > The argument, as I understand it, is that this the SENSE framework is privacy-preserving because (a) one cannot recover the original point locations and (b) one cannot recover the original inter-point distances. If I am misunderstanding, then please correct me.
> >
> > However, under the provided definition, there is still something missing. Namely, the given definition states that the x_i points cannot be uniquely determined. But this is also the case for $k=d_h$, right? Even in that setting, there are infinitely many correct solutions up to rotation. You cannot predict point locations with only distances. So why does the definition say that one needs $k < d_h$ anchors?
> >
> > Again, I understand the authors' point: by having fewer than $d_h$ anchors, the inter-point relationships are underdetermined and cannot be solved for. My original point is that it's unclear why this is necessary since one can apply an arbitrary rotation and use provably guaranteed amounts of noise to have optimally privacy-preserving embeddings. My current point is that the provided definition seems to simply be satisfied by using the distances to $k = d_h$ anchors, as the point positions are unrecoverable in that setting as well (due to distances being rotation-invariant).
> >
> > Regarding the points in W2, thank you for the clarifications. To emphasize my thoughts: the listed statements were mostly used as examples of a broader theme. In general, the paper makes statements which are not clearly substantiated or have not been introduced. For example, I am still unsure of what the authors mean that the issues are particularly problematic for tsne and umap. *All* methods compute pairwise graph distances. PCA does this, MDS does this, LLE does this, ISOMAP does this, etc. etc. Similarly, I understood what the anchors were after reading the paper. I was instead mentioning that, at this point in the introduction, the authors were using terms that had not been introduced. These are all intended as general suggestions for clarity rather than specific ones -- the authors can choose to ignore them if they disagree.

---

> > > ### Author Response · Authors · 2025-08-02
> > >
> > > We thank the reviewer for this thoughtful follow-up. Here's our clarification:
> > >
> > > 1) What We Mean by Privacy: The privacy guarantee in SENSE is not about hiding the inter-point distances in fact, these are what we aim to recover. Instead the concern is to prevent the recovery of the original high-dim features of NA data points such as patient-level attributes in healthcare datasets. These features are not mere coordinates but sensitive attribute vectors encoding medical indicators, demographics, or diagnostic measurements. Sharing or reconstructing them directly even approximately would constitute a severe privacy violation. Thus, our goal is to learn the structural similarity relationships among points across clients without revealing the sensitive high-dim features $x_i \in d_h$.  Thus, distances are the desired output while the original features are the sensitive input that must remain private.
> > >
> > > 2) Why $K < d_h$: We agree, when $K = d_h$, reconstruction is not unique it's determined up to global isometries (rotation,..etc). However, this ambiguity is limited & still allows an adversary to narrow down true point to a small, structured set. *Concrete cases:* In 2D, $K = 2$ anchors yield exactly 2 solutions: mirror images across the anchor line. In 3D, $K = 3$ anchors lie in a plane: If colinear (affinely dependent), the solution set is a full circle on a cylinder highly underdetermined. If non-colinear (affinely independent) the point lies on a unique circle (intersection of 3 spheres) still continuous but lower-dim, reducing uncertainty. With $K = 4$ coplanar anchors the feasible set collapses to 2 symmetric points reflected across the anchor plane a classic case of mirror ambiguity [1-3]. In general $\mathbb{R}^d$:  When $K = d$, the anchors span a $(d-1)$-dimensional affine subspace (hyperplane), & the feasible set lies on the intersection of hyperspheres. If anchors are affinely dependent (on a lower-dim subspace) the system remains highly underdetermined. If affinely independent, the solution lies on a narrow set possibly a low-dimensional manifold or mirror pair making recovery risk significantly higher. Thus while exact recovery isn’t always possible at $K = d$, the ambiguity is typically small structured & reversible enabling an adversary to guess or refine the true feature. In hyperbolic space exact recovery is possible with just $d$ well-placed anchors: e.g., LHYDRA [30] shows that $d$ landmarks not on a hyperplane in $\mathbb{H}^d$ suffice for recovery up to isometry.
> > >
> > > 3) Why $K < d_h$: SENSE adopts a safe general-purpose design: enforcing $K < d_h$ ensures reconstruction problem remains strictly underdetermined regardless of anchor geometry. Illustrative cases: In 2D, 1 anchor-feasible set is a circle. In 3D, 2 anchors solution lies on a spherical intersection. In $\mathbb{R}^d$, $K < d$ anchors the point lies on a high-dim manifold. These solution sets are continuous & high-entropy spread over large manifolds making exact reconstruction infeasible, even when anchor positions are known. While $K = d_h$ may yield non-unique solutions they are often structured & decodable (e.g., mirror ambiguity [1-3]). By contrast, enforcing $K < d_h$, especially the safer default $K = d_h - 1$, guarantees full underdetermination & generalizes well across dims. This choice avoids dependence on special configurations (e.g., coplanarity, affine degeneracy), making SENSE robust & privacy-preserving in practical & high-dim settings.
> > >
> > > 4) Why Not Noise-Based? The suggestion to use $K=d_h$ anchors & add noise to ensure privacy. While valid in principle this approach has practical & theoretical limitations: Noisy distances deteriorate the recovered structure especially for methods relying on geometric consistency like MDS. Since our goal is accurate inter-point distance recovery noise injection directly degrades performance. Adding noise to features (instead of distances) & then recovering distances may allow adversaries to denoise them especially if the noise distribution is known. Our goal is not DP-style privacy but structural indistinguishability ensuring that sensitive features are not reconstructible even if relative distances are learned. Limiting anchor count is natural & principled way to achieve this.
> > >
> > > 5) Clarification: We agree that all DR methods relying on global or pairwise similarities including PCA, MDS,..etc are challenged in decentralized settings where inter-client distances are unavailable. Our emphasis on t-SNE & UMAP was due to their particularly strong dependence on complete similarity graphs & negative sampling making them more fragile in such settings. We will revise the text to clarify that this limitation applies broadly & not only to NE methods.
> > >
> > > References:
> > > [1] Fang, B.T. (1986). Trilateration and GPS. J. Guid. Control Dyn.
> > > [2] Liberti, L. et al. (2014). Euclidean distance geometry & applications. SIAM Rev.
> > > [3] Biswas, P. & Ye, Y. (2004). SDP for wireless sensor localization. Proc. IPSN..

---

> > > > ### Author Response · Authors · 2025-08-04
> > > > **Rebuttal Follow-up and Request for Further Input**
> > > >
> > > > Dear Reviewer yoyK,
> > > >
> > > > Thank you again for taking the time to review our submission and for sharing your thoughtful feedback; it has been invaluable in helping us refine our work. We’ve done our best to address all the raised concerns carefully in the rebuttal, and **we’re very open to further discussion.**
> > > >
> > > > *Since we’re midway through the discussion phase*, we just wanted to gently check in to see if there are any remaining questions or points of clarification we could help with while there’s still time. *We’d be more than happy to elaborate on any aspect that might need further explanation.*
> > > >
> > > > And if you feel that your concerns have been adequately addressed, we would be sincerely grateful if you would consider updating your rating to reflect your current assessment of the work.
> > > >
> > > > Best,
> > > >
> > > > Authors

---

> > > > ### Comment · Reviewer_yoyK · 2025-08-04
> > > >
> > > > Regarding point 5, I still disagree that this is specific to tSNE and UMAP since I don't see how these models have "particularly strong dependence on similarity graphs" or how negative sampling "makes them more fragile". But it's a small detail and I don't want to get hung up on it. I was bringing it up as an example of the more general issue I find with this paper that the writing is imprecise.
> > > >
> > > > The points I'm making regarding the definition of privacy, the $K = d_h$ case, and the noise example fall into the same category. I understand what the authors are saying. I am not disputing the premise of their algorithm or its effectiveness.
> > > >
> > > > The point I am making more broadly is that the lack of precision in the authors' phrasing is an issue in the paper and, to me, continues to be an issue in the rebuttals.
> > > >
> > > > Here is a concrete example from the authors' last response. The authors say that the "concern is to prevent the recovery of the original high-dim features". Okay, so the goal of privacy preservation is that we do not want to exactly recover the positions of the points. The authors then say that, in the concrete case of $K=3$ for 3-dimensional points, the points cannot be uniquely determined and exist in a continuous space. I.e., there are an infinite number of plausible solutions. Thus, by the authors' own definition, this is privacy preserving and should be fine.
> > > >
> > > > However, the authors then say that the ambiguity is "typically small structured & reversible". How is it reversible? What is the definition of "small-structured" in the authors' perspective? I continue to try to piece apart what the authors are *implying*. To me, it feels very cut and dry: privacy-preservation implies that the exact positions cannot be recovered. The $K = d_h$ case does not allow recovering exact positions. Therefore it is privacy preserving.
> > > >
> > > > Again, I am only picking on this example because it is indicative of a larger trend. Other similar issues exist. For instance, why do the authors allow sharing 10% of the points in the multisite setting (line 239)? This seems to contradict the authors' own definition of privacy-preservation. Similarly, I don't understand why the authors are saying noise wouldn't work as it doesn't allow exactly recovering the interpoint distances. The $K < d_h$ setting they are focusing on *also* doesn't allow exactly recovering the distances!
> > > >
> > > > Furthermore, to drive the point home of what the definitional ambiguities lead to, here's yet another solution which I believe should satisfy the authors' privacy-preserving definitions. Couldn't one just apply a random rotation to all the points right at the beginning? Then their original positions are necessarily irrecoverable and one can proceed to do every downstream operation "exactly" since this doesn't corrupt the inter-point distances at all. One could even share this rotation in an encrypted way between the sites to ensure it's the same across all of them.
> > > >
> > > > To conclude, these are example concerns to highlight the broader issue. If the definitions aren't clear and the sentences are not directly supported by references or evidence, then the text allows itself to be picked apart.
> > > >
> > > > I will be keeping my score.

---

> ### Author Response · Authors · 2025-08-06
> **Responding to the remaining concerns: tsne-UMAP-**Part1****
>
> **Note: We emphasize that we are not introducing new content but thoroughly addressing your raised concerns with clearer explanations, supporting theory, citations, & empirical results. Our response is structured into 4 focused parts, each directly aligned with specific reviewer comments.**
>
> **We apologize that our previous responses did not fully resolve your concerns. We also appreciate your acknowledgment of the soundness & effectiveness of our core algorithm.** Upon reflection, we recognize that your main concern lies in the imprecise phrasing in few parts of the manuscript & we regret any resulting confusion. Below, we offer a clearer, more concise explanation:
>
> We acknowledge that all similarity-based DR methods struggle in decentralized settings without global distance access. Our phrasing may have wrongly implied this issue is unique to t-SNE/UMAP. All DR methods that depend on distances or similarities between data points face challenges when global information is unavailable. However, different methods are affected in different ways, depending on how they are designed and how they use similarity information during training. **Our emphasis on t-SNE and UMAP is based on the fact that these two methods use a different kind of optimization process, one that involves simulating forces between data points to create a meaningful low-dimensional layout.** In both t-SNE & UMAP: (Attractive force) Each point is pulled closer to points that are similar to it in the original high-dim space (typically its nearest neighbors). (Repulsive force) Each point is pushed away from other points that are dissimilar helping prevent clusters from collapsing into each other. This balance between attraction & repulsion allows t-SNE & UMAP to preserve local structure while also encouraging global separation. This force-based layout makes them powerful for visualization but also sensitive to missing information in decentralized settings. To compute repulsive forces accurately, each point would need to know about the positions of all other points in the dataset. In a centralized setting, this is possible, though computationally expensive. However, in decentralized settings, the data is spread across multiple clients & each client only has access to its own local data. This makes it impossible to directly calculate the repulsion from points on other clients, because the client has no knowledge of them.
>
> To reduce cost, t-SNE/UMAP use negative sampling: instead of computing repulsion from all points, they sample a few negatives. This works centrally, as samples reflect the global distribution. But in decentralized settings, clients can only sample from their own data a biased subset. So, the repulsive force is incomplete or skewed. This weakens separation, causes cluster drift, & distorts global structure. FEDNE (NeurIPS'24) confirms this: without accurate global repulsion, cluster layouts become unreliable. Their proposed solution uses lightweight surrogate models to approximate repulsion toward unknown global points, directly validating our concern. Similarly, the work “From t-SNE to UMAP with Contrastive Learning” (Yue et al 23) shows that t-SNE & UMAP can be understood as contrastive learning methods where the repulsion is implemented through a negative sampling-based loss. This view makes the importance of representative negatives even more central to correct optimization & also highlights why these models are particularly vulnerable when such negatives are missing or biased, as happens in decentralized settings.
>
> Other methods face different issues. PCA computes variance and finds directions of maximum spread, it does not use attraction or repulsion. MDS minimizes stress to preserve pairwise distances, but it does not use iterative force-based updates as in t-SNE. LLE & Isomap construct kNN graphs & solve global optimization problems (typically via spectral decomposition). While they rely on local neighborhoods, they still require global graph connectivity & distance information to produce meaningful embeddings. *Although these methods do not involve stochastic, runtime interactions like attractive & repulsive forces, they still fundamentally depend on access to global distances or similarities. In decentralized settings where such information is incomplete or fragmented across clients, their performance degrades significantly, though for different reasons than t-SNE or UMAP.*
>
> We thank the reviewer for reiterating about the earlier confusion. We clarified here that we do not claim t-SNE & UMAP are “more fragile” than other DR methods in general. Rather they face different challenges under decentralized conditions due to their dependence on both attractive & repulsive forces at every step of training. Their use of negative sampling to estimate repulsion & the fact that negative sampling becomes unreliable when clients can only see a small local view of the data. We will revise our manuscript to reflect this more precisely.

---

> ### Author Response · Authors · 2025-08-06
> **Responding to the remaining concerns: "typically small structured & reversible" -**Part2****
>
> In earlier responses, we used the phrase “small, structured, and reversible ambiguity” when describing the solution space under $K = D$ affinely independent anchors. We now agree that this wording was vague & potentially contradictory. Below, we clarify each term:
>
> 1) "Structured ambiguity": Although the solution is not unique, it lies on a low-dimensional, geometrically constrained manifold (e.g., a circle or mirrored points).
> 2) "Small": The space of plausible reconstructions has very low entropy (e.g., one degree of freedom), making it easier for an adversary to infer the correct solution.
> 3) "Reversible": In practice, such constrained ambiguity can often be resolved using side information, statistical priors, or optimization heuristics, as shown in localization literature and attack papers ([1–7]).
>
> So while $K = D$ doesn’t yield exact recovery, the structured ambiguity reduces effective privacy.
>
> In distance-based reconstruction (eg, DGP, localization, hyperbolic recovery) the anchor setup determines how constrained the solution space is. Privacy favors ambiguity, ideally the true point lies among many plausible candidates. Choosing $K<D$ affinely independent anchors increases uncertainty while preserving structure. This is critical for protecting sensitive attributes & holds across both geometries.
> 1) $K = D$ small, structured ambiguity: Knowing distances from $x \in \mathbb{R}^D$ to $D$ affinely independent anchors constrains $x$ to a 1D manifold (typically a circle), the intersection of $D$ hyperspheres in $\mathbb{R}^D$, yield1 DoF (rotation around the affine hull). E.g, in 2D: 2 anchors->2 symmetric positions. In 3D: 3 anchors -> point lies on a circle [8–10]. This is structured ambiguity-non-unique but tightly constrained and reversible, weakening privacy.
>
> If the $D$ anchors are affinely dependent (e.g., lie on a hyperplane), the system degenerates & the solution expands from a curve to a surface or higher. Eg: In 2D: 2 collinear anchors-> infinite feasible points on a circle. In 3D: 3 coplanar anchors -> solution on a cylindrical surface (2D ambiguity) [9]. While ambiguity aids privacy, affine dependence is fragile, hard to detect, & unreliable in high dimensions.
>
> 2) $K<D$ robust, privacy-safe ambiguity: When the #anchors is reduced to $K = D - 1$ & they are affinely independent, the intersection of $D - 1$ hyperspheres in $\mathbb{R}^D$ leaves the unknown point on a 1D manifold (if $K = D - 1$) or a 2D manifold (if $K = D - 2$). This increases ambiguity while preserving structure & analyzability. Eg: In 3D: 2 anchors->solution on sphere surface (2D ambiguity). In 4D: 3 anchors-> 2D manifold in $\mathbb{R}^4$. ([9], Ch. 3.2). *This ambiguity is affine-robust even poorly placed anchors preserve uncertainty.* Moreover, L-HYDRA shows recovery is possible when $K \geq d$ (Thm 3.1), so $K = D-1$ is the last point before identifiability arises.
>
> Why $K<D$ is a Privacy-Safe Design Choice: More Ambiguity = More Privacy: Fewer constraints expand the solution space (e.g., curve -> surface), making inference harder. Affine-Robust: Unlike $K = D$, no risk of degeneracy from anchor placement. No Utility Loss: Most applications (e.g., clustering, visualization) only need recovery up to isometry; more anchors add identifiability risk [1-6] without added benefit. **Thus, choosing $K < d$ affinely independent anchors is a principled privacy strategy it maximizes ambiguity, avoids degeneracy & generalizes across both spaces, making it broadly applicable. In contrast, $K = D$ imposes tight constraints & narrows the solution space. SENSE intentionally limits anchor access to protect against this.**
>
> **NOTE1:** Mirror ambiguity [10-16] occurs when localization via anchor distances yields multiple symmetric solutions, especially at $k = d$ (eg, 2 anchors in 2D). [10] studies this in 3D WSNs where flip ambiguity causes 2 equally plausible mirrored node positions due to geometry or noise. Why it happens: In 2D with $k = 2$, the 2 anchor distances define circles intersecting at 2 symmetric points, reflections across the anchor line. Symmetric distances: Knowing a distance, the point may lie on either side of the anchor. In [16] localization is framed as embedding a weighted graph (nodes = unknowns + anchors, edges = distances) into $\mathbb{R}^k$. With few anchors, especially $k = D$, mirror symmetry arises: multiple equally valid placements of unknown nodes exist, reflected across the affine subspace spanned by the anchors.
>
> Ref (abbreviated):
> [1] Fredrikson et al. (CCS’15) · [2] Shokri et al. (S&P’17) · [3] Fredrikson et al. (USENIX’14) · [4] Zhu et al. (NeurIPS’19) · [5] Geiping et al. (NeurIPS’20) · [6] Nasr et al. (S&P’19) · [7] Fang (J. Guid. Ctrl Dyn’86) · [8] Liberti et al. (SIAM Rev’14) · [9] Biswas & Ye (IPSN’04) · [10] Liu et al. (ICT’14) · [11] Bose et al. (arXiv’17) · [12] Hou (GPS Solut.’22) · [13] Gerok et al. (ICUWB’09) · [14] Betti et al. (Geo’93) · [15] Teunissen (GNSS’17) · [16] Saxe (Allerton’79)

---

> ### Author Response · Authors · 2025-08-06
> **Responding to the remaining concerns: Clarifying Anchor Sharing and Noise-Based Privacy-**Part3****
>
> We sincerely thank the reviewer for identifying this issue, and we agree that our earlier phrasing may not have made the distinctions clear. Below we address both subpoints in turn.
>
> 1) Anchor Sharing Does Not Violate Our Privacy Definition: The 10% anchor sharing in the multisite setting (line 239) is used only for empirical evaluation. These anchors are not private they act as public or semi-public landmarks, similar to those in GPS or radar systems. This is standard practice in localization literature [6,7], where landmarks aid positioning but are not themselves privacy-sensitive [8]. Our privacy definition protects only the high-dimensional features of NA points. Anchors are fixed & visible and either synthetic, public, or explicitly consented. Not part of any client's private data. We do not rely on anchor secrecy but instead limit anchor quantity, ensuring $K < d$ to maintain ambiguity.
>
> *Real-World Anchor Sources:* In practical deployments, anchors are selected based on domain knowledge and are not sampled arbitrarily from private data. Eg, Healthcare: Publicly released reference scans or patient-consented samples [1,2]. Finance: Standard transaction patterns or aggregated customer profiles [3]. Genomics: Population-level reference genomes (e.g., 1000 Genomes, UK Biobank) [4,5]. Synthetic Anchors: Via MMD minimization [5], though limited in coverage and potentially adversarial (see also Reviewer yoyK Q4 and rkR3 W2, Table 1). Trusted server-curated anchors are auditable, robust, and independent of client records, reducing leakage risks like membership inference [5]
>
> This design is further supported by theory on low-rank recovery via anchor distances [9], matrix completion in non-orthogonal bases [10], and Gram matrix-based localization [11].
>
> 2) Why SENSE Avoids Noise-Based Privacy: We thank the reviewer for raising the important question of noise-based privacy mechanisms. While we acknowledge that such approaches can be used, the goal of SENSE is fundamentally different. Our objective is to reconstruct or approximate inter-point distances not the raw features while ensuring that high-dim features of NA points remain private. Importantly in SENSE distances are not treated as private. Instead, they are intermediate coordination variables similar to how they are used in classical MDS or sensor network localization. The core idea is to protect the original feature representations, not to hide the distance map itself. That said, we recognize that noise-based privacy mechanisms may be appropriate in some scenarios, particularly where (a) robust denoising resistance is guaranteed & (b) added noise does not critically degrade performance. However, we chose not to use noise-based methods in SENSE for the following reasons (also see Reviewer yoyK ans to Q3):
>
> Empirical Evaluation: 1) Noise Added to DNA Matrix: We randomly selected a percentage of clients & added noise to their client-to-anchor distances. Evaluation on datasets (Iris & Seeds) shows a clear trade-off between privacy & performance:
> |Data|Metric|0%|5%|10%|20%|50%|
> |-|-|-|-|-|-|-|
> |Iris|FS|0.8659|0.8403|0.8104|0.7000|0.6167 |
> ||DE|0.0425|0.0735|0.0929|0.1528|0.1828|
> |Seeds|FS|0.9745|0.9390|0.9038|0.7902|0.6589|
> ||DE|0.0102|0.1251|0.1760|0.2471|0.3702|
>
> As noise increases (that is, increasing the noisy entries in DNA block by inc noisy clients percentage), both FS drops & DE rises, demonstrating that such noise harms utility in SENSE.
>
> 2) Noise to Raw Features: We also added Gaussian noise directly to raw features (e.g., pixel-level perturbation of MNIST data) with $K = d_h$ anchors & a client set of 10 clients & 100 NA. In the noise-based variant (NS), we added element-wise zero-mean Gaussian noise with varying standard deviations proportional to data range:
> |Method|DE|FS|
> |-|-|-|
> |SENSE|0.006023|0.974955|
> |NS1|0.074474|0.946917|
> |NS2|0.268156|0.887382|
> |NS3|0.535183|0.801178|
> |NS4|0.843552|0.691761|
> |NS5|1.175783|0.573909|
>
> **These results show that even mild noise causes sharp performance degradation, confirming that noise-based privacy comes at a significant cost to utility (comparing SENSE without noise with NS). In real-world applications such as healthcare or finance, where precision is critical, this trade-off is often unacceptable.**
>
> References:
> [1] Bycroft et al. (Nature’18). UK Biobank, Genomics.
> [2] Johnson et al. (SciData’16). MIMIC-III Dataset.
> [3] Awosika et al. (Access’24). XAI & FL in Finance.
> [4] Litvinuková et al. (Nature’20). Human Heart Atlas.
> [5] Qiao et al. (2024). Fed t-SNE & UMAP.
> [6] Patil & Zaveri (WSN’11). Trilateration via MDS.
> [7] Fang & Chen (Sensors’20). RSS-Based Trilateration.
> [8] Nikoo & Behnia (TAES’18). Passive Localization.
> [9] Lichtenberg & Tasissa (TIT’24). Low-Rank Recovery.
> [10] Tasissa & Lai (LAA’21). Matrix Completion (Non-Orth. Basis).
> [11] Kundu et al. (CISS’25). Sampling for Distance Geometry.
> [12] Xu et al. (TOG’19). Adversarial Denoising (AMC).
> [13] Tan et al. (arXiv’25). Attribution in Gen Models.

---

> ### Author Response · Authors · 2025-08-06
> **Responding to the remaining concerns: "Random rotation to all the points right at the beginning"-**Part4****
>
> We thank the reviewer for this suggestion; as per our understanding, a global rotation does not provide privacy in a multi-client setting. It is only effective if applied to the entire dataset, which is infeasible in decentralized settings where clients don’t share raw features or a common coordinate frame. Moreover, if the rotation is known (shared in encrypted form), then it may be inverted using existing techniques (discussed in points 1-3). If it’s unknown, coordination between clients becomes impossible. SENSE avoids both issues by never assuming access to global feature vectors; privacy arises from the geometric ambiguity of the reconstruction problem itself, not from post-hoc obfuscation.
>
> 1) Image encryption and decryption using bit rotation and Hill Cipher: A paper discusses an encryption approach for images using bit rotation reversal and extended Hill cipher techniques. Decryption involves reversing the bit rotation and applying the inverse Hill cipher [1]
> 2) Vertical Bit Rotation (VBR) Algorithm: The decryption process for this algorithm involves applying the opposite rotation (e.g., rotating vertically upwards if encryption rotated downwards) using the same key to restore the original data [2]
> 3) Permutation based Image Encryption Technique: This technique involves multiple phases including bit, pixel, and block permutation. Decryption reverses these permutation steps in the correct order to reveal the secret image [3]
>
> We also acknowledge that every privacy mechanism has its strengths and limitations. While noise and rotation-based methods offer certain guarantees, SENSE takes a different approach, grounded in deterministic geometry and suitable for decentralized, structured, and privacy-sensitive scenarios. We see this as a complementary direction in the space of privacy-preserving representation learning. We will add a remark in the revised manuscript to explicitly discuss: Future directions, including potential hybrid methods that combine SENSE with noise-based or rotation-based techniques. Opportunities to strengthen SENSE's privacy guarantees using additional regularization or uncertainty modeling. We sincerely thank the reviewer for these insightful suggestions and would be happy to explore them further in future work.
>
> References:
> [1] Kumar et al. (IJCA’12). Bit Rotation & Hill Cipher.
> [2] Pertiwi et al. (JoSCE’22). Vertical Bit Rotation Algorithm.
> [3] Indrakanti & Avadhani (IJCA). Permutation-Based Image Encryption.
>
> We agree that our writing in some parts conflates different types of ambiguity (e.g., “non-uniqueness” vs “reversibility”). To address this, we will: 1) Add a formal definition of privacy in the introduction and make the introduction more clear by clarifying how the NE and DR methods suffer in decentralized settings. 2) Clarify why we are choosing $K<d$ and not $K = d$. 3) Remove confusing language like “small-structured & reversible” unless we define it precisely with illustrative examples (e.g., 3D circle on sphere intersection, 2D mirror reflection). 4) Clearly distinguish between anchors (non-sensitive) and private points, justifying anchor sharing in practice and theory. 5) Emphasize why post-hoc rotation or noise is insufficient in decentralized settings without raw feature sharing or coordination.
>
> **Appeal to Reviewer:**
> We sincerely thank the reviewer for pushing us to sharpen our definitions and improve the clarity of our manuscript. We hope that the combination of empirical evidence, theoretical grounding, and clearer articulation addresses the key concerns raised. As all other reviewers are leaning toward acceptance, your feedback is critical in determining the final outcome of this work. If you feel your concerns have now been sufficiently addressed, we would deeply appreciate if you could consider revisiting your overall score to reflect your updated assessment. We are fully committed to incorporating all your suggestions in the final revision.

---

> > ### Author Response · Authors · 2025-08-08
> >
> > **Appeal to Reviewer yoyK:** We sincerely thank the reviewer for pushing us to sharpen our definitions and improve the clarity of our manuscript. We hope that the combination of empirical evidence, theoretical grounding, and clearer articulation addresses the key concerns raised. As all other reviewers are leaning toward acceptance and also the discussion phase is ending, your feedback is critical in determining the final outcome of this work. If you feel your concerns have now been sufficiently addressed, we would deeply appreciate if you could consider revisiting your overall score to reflect your updated assessment. We are fully committed to incorporating all your suggestions in the final revision.

---

> ### Comment · Reviewer_yoyK · 2025-08-08
>
> I sincerely appreciate and understand the authors' efforts in the response. Really, I do.
>
> But, to re-iterate, my point is not in these *specific* concerns. I used these as examples of a more general trend in the paper. My primary issue is that the writing is consistently imprecise (makes unsubstantiated claims, lacks definitions, does not adequately express why algorithmic decisions are reached). These issues I brought up were examples of this.
>
> While I recognize the authors' insistence regarding the examples I presented (I still disagree but this is not important), my primary concern is in the overall issue rather than the points in question.
>
> I recognize that the authors state that they will resolve this for the camera-ready, but as I am not able to see what these revisions would be, I do not feel confident that the *thematic* issues will be resolved (in particular because the authors' first comments produced a definition of privacy-preserving that then led to further contradictions and imprecisions).
>
> I believe the authors are presenting a good idea. I believe that when it is re-written so that every statement is checked/cited/substantiated, that it will be a great paper. But in the form it was presented, I do not believe it passes the bar for NeurIPS. I am sorry. As I said, I will be keeping my score.

---

> > ### Author Response · Authors · 2025-08-09
> >
> > Thank you for actively engaging in the discussion and for your constructive feedback. This exchange has been valuable in helping us clarify your concerns about the core ideas of the paper. Your suggestions, especially regarding the definition of privacy, have helped us see how we can improve clarity without altering the fundamental flow of the work. We fully agree that precision in presentation is important. The suggested phrasing improvements are straightforward and can be readily integrated into the manuscript. If we had the opportunity to update the submission during the discussion phase, we would have already incorporated these refinements without changing the substance of the work. If given the chance in the revision stage, we will certainly integrate all your suggestions, such as:
> >
> > - *Providing a clear and formal definition of privacy.*
> > - *Clarifying why choosing $K < d$ is a better generic choice than $K = d$* or adding small amounts of noise, as supported by our empirical study and well-cited arguments (see Responding to the remaining concerns: Clarifying Anchor Sharing and Noise-Based Privacy — Part 3).
> > - *Ensuring the introduction’s explanation of t-SNE/UMAP challenges in decentralized settings* is as clear as in our rebuttal (Responding to the remaining concerns: t-SNE–UMAP — Part 1).
> >
> > We are grateful for your recognition of the work’s methodological thoroughness, breadth of experiments, careful presentation of related work, and the soundness of the approach. We also appreciate your encouragement that, with each statement clearly substantiated, the paper can become much stronger. These refinements can be incorporated smoothly, and we are fully committed to doing so in the camera-ready version.
> >
> > We assure you that these presentation refinements can be made smoothly, and we are fully committed to integrating them into the revised manuscript.
> >
> > *To ensure our position is clear:* we believe the substantive scientific aspects have been addressed, and the remaining points relate to presentation refinements, which we will incorporate with care in the camera-ready version.
> >
> > Best,
> >
> > Authors

---

### Author Response · Authors · 2025-08-09
**Summary of Rebuttal:**

We sincerely thank the reviewers for their constructive and thoughtful feedback. Your comments have helped us improve both the clarity of our presentation and the depth of our experimental and theoretical validation. Below is a consolidated summary of how we addressed the main concerns across reviewers, along with the new experiments and clarifications provided.


### **1. Runtime, Scalability, and Theoretical Support:**
- *Runtime analysis*: Experiments comparing SENSE with SOTA on both accuracy & full-pipeline runtime analysis (anchor generation → matrix completion → t-SNE embedding).
- *Pipeline breakdown*: Reported runtime for each SENSE stage & provided complexity analysis.
- *Scalability:* Conducted experiments on SVHN (80K non-anchors) & Tiny ImageNet (90K non-anchors).
- *Theoretical support*: Clarified privacy & fidelity guarantees, including robustness against structure-aware adversaries via empirical attack simulations.


### **2. Privacy, Adversary Models & Anchor Effects:**
- *Three-fold privacy validation*: 1) Theoretical non-invertibility when $K < d$. 2) Clarification on incompatibility of Standard Model Inversion & Membership Inference Attacks in SENSE. 3) Empirical simulations on 4 datasets.
- *Privacy–utility–compute trade-off*: Empirically demonstrated effects of anchor count ($K$) & geometry on fidelity and runtime.
- *Anchor selection & geometry*: Experiments to show that in-distribution anchors outperform random sampling and also that affinely independent anchors improve reconstruction.
- *Practical anchor sourcing*: Discussed realistic, non-sensitive anchor choices for decentralized setups.


### **3. Definition of Privacy, Presentation Precision, and Metrics:**
- *Formal privacy definition*: Clarified why $K < d$ is a deliberate design choice & why $K = d$ with added noise is not in our setting (details & empirical evidence in "Responding to the Remaining Concerns: Clarifying Anchor Sharing and Noise-Based Privacy – **Part 3**").
- *Metric justification*: Explained the use of Trust., Cont., Stead., & Cohes. which are well-established in recent DR-federated visualization work. Added experiments with CA, NMI, SC, & Overlap@k for benchmarking & showed how SOTA methods are more vulnerable to adversaries (A1, A2) compared to SENSE.
- Added curated dataset experiments showing that SENSE embeddings do not reveal sensitive details.
- *Clarifications in NE context*: Clarified the confusion on MDS from modern NE methods like t-SNE/UMAP. Provided a detailed explanation of t-SNE/UMAP issues in decentralized settings (details in "Responding to the remaining concerns: tsne-UMAP-**Part1**"), with supporting citations.
  - Clarified the link between NE and contrastive learning.
  - Defined anchors as non-sensitive reference points & explained the rationale for anchor sharing (details in "Responding to the remaining concerns: Clarifying Anchor Sharing and Noise-Based Privacy-**Part3**").
  - Addressed concerns about applying random rotations (details in "Responding to the remaining concerns: "Random rotation to all the points right at the beginning"-**Part4**").
  - Clarified terms such as “typically small, structured & reversible” ("Responding to the remaining concerns: "typically small structured & reversible" -**Part2**").


## **Acknowledgement**
By the end of discussions, most reviewers agreed that their core concerns had been satisfactorily addressed. Reviewer yoyK’s remaining points focus on presentation precision, refinements in phrasing, definitions, & organization, which are not actionable at the rebuttal stage but will be implemented in the camera-ready version. Specifically, we will: Formally define privacy in the introduction. Clarify the $K < d$ design choice and why adding noise is not equivalent. Refine the t-SNE/UMAP discussion for clarity. Incorporate all suggested phrasing improvements.

We appreciate the Area Chair’s guidance in ensuring a balanced and productive review process. All clarifications, experiments, and theoretical additions discussed during rebuttal will be fully integrated into the final manuscript.

**Best regards,**
*Authors*

---

### Note · Authors · 2025-08-12

We sincerely thank the AC and reviewers for their time and constructive feedback. Your comments have helped us improve both the clarity of our presentation and the depth of our experimental and theoretical validation. **We will ensure that all the clarifications, additions, and improvements discussed during the rebuttal are carefully incorporated into the revised manuscript.**

As detailed in our **“Summary of Rebuttal,”** we have addressed all substantive concerns with new experiments, theoretical clarifications, and extended empirical validation.

**Experimental side:** We performed additional large-scale evaluations on datasets such as SVHN and Tiny-ImageNet, including detailed stage-wise runtime and complexity analysis to confirm scalability. We expanded privacy evaluations to cover adversaries such as model inversion, membership inference, and structure-aware attacks, showing that the framework preserves privacy while maintaining high fidelity under diverse conditions. We also compared noise-based and anchor-based privacy, demonstrating through quantitative metrics that noise significantly reduces utility.

**Theoretical side:** We formalized the privacy definition with a focus on protecting high-dimensional private features, supported by literature in distance geometry and network localization. We justified why $K<d$ is a more generic and robust design choice than $K=d$ or noise-based methods, illustrating the “small structured & reversible” ambiguity with concrete geometric examples and references. We clarified the confusion regarding t-SNE and UMAP senstivity in decentralized settings along with other DR methods and also discussed the reliance of tsne and umap on biased force-based layouts and negative sampling. We also provided well-cited evidence explaining the role of public/semi-public/synthetic anchors as non-sensitive reference points that enable geometric coordination without compromising privacy, along with practical examples of how such anchors can be sourced in decentralized setups.

The remaining suggestions relate to writing and clarity of phrasing. These are straightforward editorial refinements that will be fully integrated into the revised version without altering the substance of the work.

---

### Decision · Program_Chairs · 2025-09-17

**Decision:**

Reject

**Comment:**

This paper received mixed reviews: out of the three received reviews, one recommended acceptance [7okj], one recommended borderline acceptance [rkR3] and one recommended rejection [yoyK].

Reviewers appreciated some aspects of the work:

+ The thoroughness of the methodology was appreciated [yoyK]
+ The theoretical foundation was considered solid [7okj] and was appreciated [rkR3]
+ The strategy was considered creative and interesting [rkR3]
+ The method was considered sound [yoyK]
+ The approach was considered useful for downstream applications [7okj]
+ Computational efficiency of the method was appreciated [7okj]
+ The large number of datasets was appreciated [yoyK]
+ The selection of neighbour embedding methods was appreciated [yoyK]
+ Good experimental performance was appreciated [7ojk]
+ The ablation studies were appreciated [yoyK]

However, the reviewers also raised several concerns, which led to extensive discussion in the rebuttal stage. In particular there was discussion regarding the meaning of the privacy definition and its implications. In more detail:

- The impact of the problem (its existence in the community) was considered unclear [yoyK]; authors argue there is recent work on privacy-preserving NE.
- Some claims were considered exaggerated such as bounds on reconstruction fidelity, and guarantees using projection-aware SMACOF were desired [rkR3]; authors agreed they did not provide fidelity bounds but argued they were established in literature, and argued their "fidelity" was about neighbourhood structure instead of feature recovery.
- Lack of scalability demonstration to large datasets was criticised and a runtime analysis was desired [rkR3]; authors provided a runtime table showing better performance than FedTSNE.
- Lack of a proper privacy definition was criticised [yoyK] and it was considered unclear why the authors' situation was considered privacy preserving despite large agreement with original distances; essentially there was concern that the method is too easily called privacy preserving [yoyK]. Author and the reviewer carried out a lengthy argumentation, where among other things, authors provided one definition (arguing that underdetermined systems of unknowns would make exact recovery impossible). Authors also provided a simple example of a SENSE-embedded data arguing it does not expose sensitive details; and later noted that anchor sharing would not violate their privacy definition. (For brevity, not all of the argumentation is summarised here.)
- Related to the above discussion, it was felt the distance may reveal some information about the features [7okj], and it was questioned whether non-exact recovery is enough for privacy [rkR3]; reviewers argued the system is undetermined and would not recover features well, and thus standard privacy attacks would not work. Authors also provided an empirical attack simulation.
- It was questioned what would happen if the data lived in a low-dimensional subspace or was sparse [rkR3]; authors provided some discussion and some proposed solutions.
- Performance evaluation on more sophisticated datasets was desired [7okj]; authors provided results on three more datasets.
- Better ablation of the anchor selection criteria was desired, and discussion of how qualitatively and theoretically anchor selection strategy affects privacy robustness [7okj]; authors argued their strategy has optimality properties, and provided some results on random anchor sampling versus in-distribution anchors; they also discussed practical sources of anchors.
- Considering simple noise addition for privacy was suggested [yoyK]; authors noted it would not suffice without formal guarantees and would be fragile whereas their geometric non-identifiability would be stronger. They also provided one experiment on noise based privacy.
- Lack of clarity in several parts of the presentation was criticised [yoyK]
- The number of selected anchors was unclear [rkR3]
- It was questioned whether the two stages could be merged [rkR3]; authors stated this is a direction they are exploring.
- An anchor based method was requested for background [7pkj]; authors planned to provide brief description.
- Clarifying the metrics used and why they are natural choices instead of more standard ones was requested [yoyK]; authors provided some discussion of the metrics, and provided a table of results for some other metrics.
- Better discussion of limitations was desired [yoyK]


After the rebuttal stage, despite extensive discussion, reviewer [yoyK] remained concerned about imprecise writing and lack of rigour in the claims. Reviewer [7okj] considered the core idea interesting and well evaluated but still found the clarity lacking in the early paper. Reviewer [rkR3] considered the questions and concerns addressed.


Overall, to me the extensive discussion shows the clarity of the proposed ideas is not yet communicated well enough in the current work. It remains somewhat unclear how strong the remains concerns of [yoyK] ultimately are but also [7okj] found clarity lacking in the early paper.  Essentially, it seems that central aspects of the motivation have been clarified during the rebuttal stage, but it remains unclear how clear these would become to a reader of the final paper. As NeurIPS is a highly competitive venue, ultimately I feel that even though the idea is quite interesting as a new direction in privacy preserving visualisation and embedding, in its current state the work may unfortunately not yet be ready to be presented at NeurIPS. I encourage the authors to incorporate the proposed changes and clarify and crystallize the argumentation, with such changes I believe the work has a good chance to be acceptable in another venue.